# Proteomic analysis of the urothelial cancer landscape

Franz F. Dressler [1,2,3] ✉, Falk Diedrichs [1], Deema Sabtan[1], Sofie Hinrichs[3], Christoph Krisp [4], Timo Gemoll[5], Martin Hennig[6], Paulina Mackedanz[3], Mareile Schlotfeldt [3], Hannah Voß[4], Anne Offermann[3], Jutta Kirfel[3], Marie C. Roesch[6], Julian P. Struck[6,7], Mario W. Kramer[6], Axel S. Merseburger[6], Christian Gratzke[8], Dominik S. Schoeb[8], Arkadiusz Miernik [8], Hartmut Schlüter [4], Ulrich Wetterauer[8,9], Roman Zubarev [10,11,12], Sven Perner[3,13,14], Philipp Wolf [8] & Ákos Végvári [10]

Urothelial bladder cancer (UC) has a wide tumor biological spectrum with challenging prognostic stratification and relevant therapy-associated morbidity. Most molecular classifications relate only indirectly to the therapeutically relevant protein level. We improve the pre-analytics of clinical samples for proteome analyses and characterize a cohort of 434 samples with 242 tumors and 192 paired normal mucosae covering the full range of UC. We evaluate sample-wise tumor specificity and rank biomarkers by target relevance. We identify robust proteomic subtypes with prognostic information independent from histopathological groups. In silico drug prediction suggests efficacy of several compounds hitherto not in clinical use. Both in silico and in vitro data indicate predictive value of the proteomic clusters for these drugs. We underline that proteomics is relevant for personalized oncology and provide abundance and tumor specificity data for a large part of the UC proteome (www.cancerproteins.org).

Urothelial bladder cancer (UC) challenges diagnostic and clinical management with a wide tumor biological spectrum and frequent progression from non-invasive to life-threatening advanced cancer. At the same time follow-up monitoring of this common tumor[1,2] requires transurethral examinations of the bladder, leading to further risks, discomfort, and high costs[3]. Radical surgery has an immense functional impact and can cause additional non-cancer mortality and morbidity[4]. Bladder-preserving therapies also have considerable side effects and are of limited efficacy[5–7]. Optimal treatment of UC thus relies heavily on both prognostic risk stratification and prediction of individual therapy response[8].

[1]Institute of Pathology, Charité - Universitätsmedizin Berlin, corporate member of Freie Universität Berlin, Humboldt-Universität zu Berlin, and Berlin Institute of Health, Berlin, Germany. [2]Berlin Institute of Health at Charité - Universitätsmedizin Berlin, Berlin, Germany. [3]Institute of Pathology, University Medical Center Schleswig-Holstein, Campus Lübeck, Lübeck, Germany. [4]Section Mass Spectrometry and Proteomics, Campus Forschung N27 00.008, University Medical Center Hamburg-Eppendorf, Hamburg, Germany. [5]Section for Translational Surgical Oncology and Biobanking, Department of Surgery, University Medical Center Schleswig-Holstein, Campus Lübeck, Lübeck, Germany. [6]Department of Urology, University Hospital Schleswig-Holstein, Campus Lübeck, Lübeck, Germany. [7]Department of Urology, Faculty of Health Sciences Brandenburg, Brandenburg Medical School Theodor Fontane, Brandenburg, Germany. [8]Department of Urology, Medical Center—University of Freiburg, Faculty of Medicine, University of Freiburg, Freiburg, Germany. [9]Department of Medicine, Faculty of Medicine and Dentistry, Danube Private University, 3500 Krems, Austria. [10]Division of Chemistry I, Department of Medical Biochemistry and Biophysics, Karolinska Institutet, Stockholm, Sweden. [11]The National Medical Research Center for Endocrinology, Moscow, Russia. [12]Department of Pharmacological & Technological Chemistry, I.M. Sechenov First Moscow State Medical University, Moscow, Russia. [13]Institute of Pathology, Research Center Borstel, Leibniz Lung Center, Borstel, Germany. [14]Center for Precision Oncology, Tuebingen, Germany. ✉e-mail: franz-friedrich.dressler@charite.de

With the availability of high-throughput micro-arrays and RNA sequencing several transcriptome-based classification schemes have been proposed[9–18]. These cohorts focused mostly on advanced, muscle-invasive urothelial bladder cancer (MIBC). Balancing risks and treatment, however, is even more difficult in the earlier stages of non- and superficially invasive UC, for which radical surgery is an option, not the rule. Clinical key questions of personalized risk stratification thus remain unsolved.

Another long-standing goal in clinical UC management is the optimal use of adjuvant therapies. Recently, surface protein-specific antibody-drug conjugates (ADC) have been developed[19–21]. While some of these therapeutics have advanced to first-line clinical application with exceptional response rates[22,23], data on the actual tumor specificity of their targets is sparse and based on semiquantitative methods such as immunohistochemistry (IHC)[24–28]. Transcriptomic analyses allow only indirect measurement of protein abundances and are quantitatively insufficient. The assumption of these target proteins being universally expressed, yet simultaneously tumorspecific, remains largely untested and unproven.

To address these ever-relevant issues of both prognostic and predictive molecular personalization, we turn to the proteome, where the vast majority of cellular effector molecules is found. While shortening the distance from descriptive association to targetable molecular function, proteomic assays are non-amplifying and pose several experimental hurdles based on the irretrievable loss of analytes. Especially routine diagnostic formalin-fixed paraffin-embedded (FFPE) tissue adds further complexity and reduces the analytical depth due to heterogeneous processing and fixation artifacts.

In this work, we further advance pre-analytical sample processing to increase the analysis depth of clinical FFPE specimens and to improve robustness across setups and the heterogeneous samples of the UC landscape. We use liquid chromatography-coupled tandem mass spectrometry (LC-MS/MS) to analyze a comprehensive cohort covering the entire UC spectrum including pairs of healthy and tumor tissue. Our analysis reveals prognostic and predictive molecular patterns and yields quantitative data on the actual tumor specificity of various biomarkers on the proteomic scale. We validate our findings by orthogonal proteomic methods, contextualize our results within transcriptomic classification schemes and provide in vitro evidence of the biological and predictive relevance of proteomic subtypes.

## Results

To analyze a comprehensive all-stage UC cohort from routine FFPE tissue and to measure protein target specificity quantitatively, we developed an optimized protocol to increase the analysis depth for standard clinical samples with heterogeneous tissue types, fixation, and processing.

### Surface maximization and reduction of dimensionality (SMRD) increases analysis depth

To cover the clinical UC spectrum, sufficient performance, even with small amounts of input material, needs to be ensured, which is typically obtained after minute dissection, e.g., for focally invasive areas. In addition, analysis depth is particularly relevant as key oncogenic proteins are of comparably low abundance and are often transmembrane proteins. Liquid chromatography-coupled tandem mass spectrometry (LC-MS/MS) of complex biological samples is negatively affected by quantitative imbalances within the sample. These are caused by highly abundant proteins such as smooth muscle or extracellular matrix (ECM) proteins. Membrane proteins, in turn, are less accessible because of their reduced solubility, co-localization with structural and ECM proteins, and geometric abundance imbalance (cytosolic proteins covering the cell volume exceed cell surface proteins by one order of magnitude, linearly biased by cell size). We enzymatically and mechanically disassembled the ECM (freeing the cell surface) and thermally disrupted the cells to produce cell fragments. FFPE samples are chemically glued together by the fixative, which yielded pieces of cell membrane and organelles with attached cytosolic proteins (Supplementary Fig. 1). We pelleted these fragments, solubilized the proteins, and included both solubilized proteins and fragments with residual (membrane) proteins for direct trypsinization (Fig. 1a, b).

We evaluated the performance of SMRD against direct trypsinization in a controlled setting and used different LC-MS/MS setups and quantitation methods to ensure maximal robustness. SMRD increased membrane protein intensities in a proteomic experiment of tandem mass tag-labelled multiplexed samples (TMT; Fig. 1c). We chose a highly specific collagenase, which did not increase unspecific protein degradation compared to controls—in contrast to commercially available tissue dissociation mixes, which led to nearly twice as many degraded peptides (Fig. 1d). Using the example of four UC and one non-small cell lung cancer specimens, we tested our protocol in label-free, single-run experiments with two different LC-MS/MS setups. Our approach raised the number of all identified proteins by mean 30%, with a higher increase in membrane proteins (Fig. 1f–h).

Additional enzymatic deglycosylation (to facilitate the identification of membrane proteins, which are frequently glycosylated) raised the number of proteins in some replicates (Fig. 1f–h; medium red bars), but the increased sample handling time and costs, as well as the potential variance source by another enzymatic reaction, prevented application for all samples in the study cohort. Further membrane protein enrichment strategies were attempted but proved unsuccessful or not scalable to cohort analyses. Supplementary Note 1 reports these negative results[29–33].

### Cohort characteristics

To ensure an unbiased clinical cohort, a continuous series of 877 UC cases of the urinary bladder, including all UC submissions over a five-year period, was reviewed. After exclusion of external patients without clinical data, 627 cases were reviewed pathologically. 271 cases had sufficient material and matching healthy mucosa from the same case. After SMRD processing, peptide concentrations were sufficient in 84% of all samples. 435 samples were submitted to labelling and LC-MS/MS analysis, comprising 242 tumor samples from 196 patients and paired healthy tissue of 193 cases. 434 samples (99.7%) demonstrated sufficient peptide intensities for quantification, with one healthy sample being excluded from further analysis, leading to n = 192 sample pairs (CONSORT diagram in Supplementary Fig. 2).

The samples covered all stages of UC (Supplementary Table 1): non-invasive papillary (pTa) low-grade, n = 38; pTa high-grade, n = 55; carcinoma in situ (pTis), n = 14; stroma-invasive (pT1) up to the lamina muscularis mucosae (pT1a/b) and up to medium grade, n = 34; deep stroma-invasive across the lamina muscularis mucosae (pT1c) and/or high-grade (G3), n = 26; muscle-invasive (pT2), n = 50; >pT2, n = 22; with lymph node metastasis (pN1), n = 3. Together, n = 167 samples were non-muscle-invasive (NMIBC; 69%) and n = 75 samples were muscle-invasive (MIBC; 31%) bladder cancer specimens. Based on availability, we included two (n = 34 patients) or three (n = 6) metachronous samples from individual patients to study temporal effects/ stability. In line with the UC gender prevalence four out of five patients were male. For surviving patients complete five-year overall survival data were available for 93% with median 7.9 years follow-up. For recurrence-free and progression-free survival follow-up was available in 86% and 89% with median 35.5 and 25.0 months, and complete two-year follow-up for 58% and 51% respectively. 76% of the samples were transurethral resections, the rest were cystectomy specimens. 77% of the samples had not received prior instillation therapy, 97% were chemotherapy-naïve.

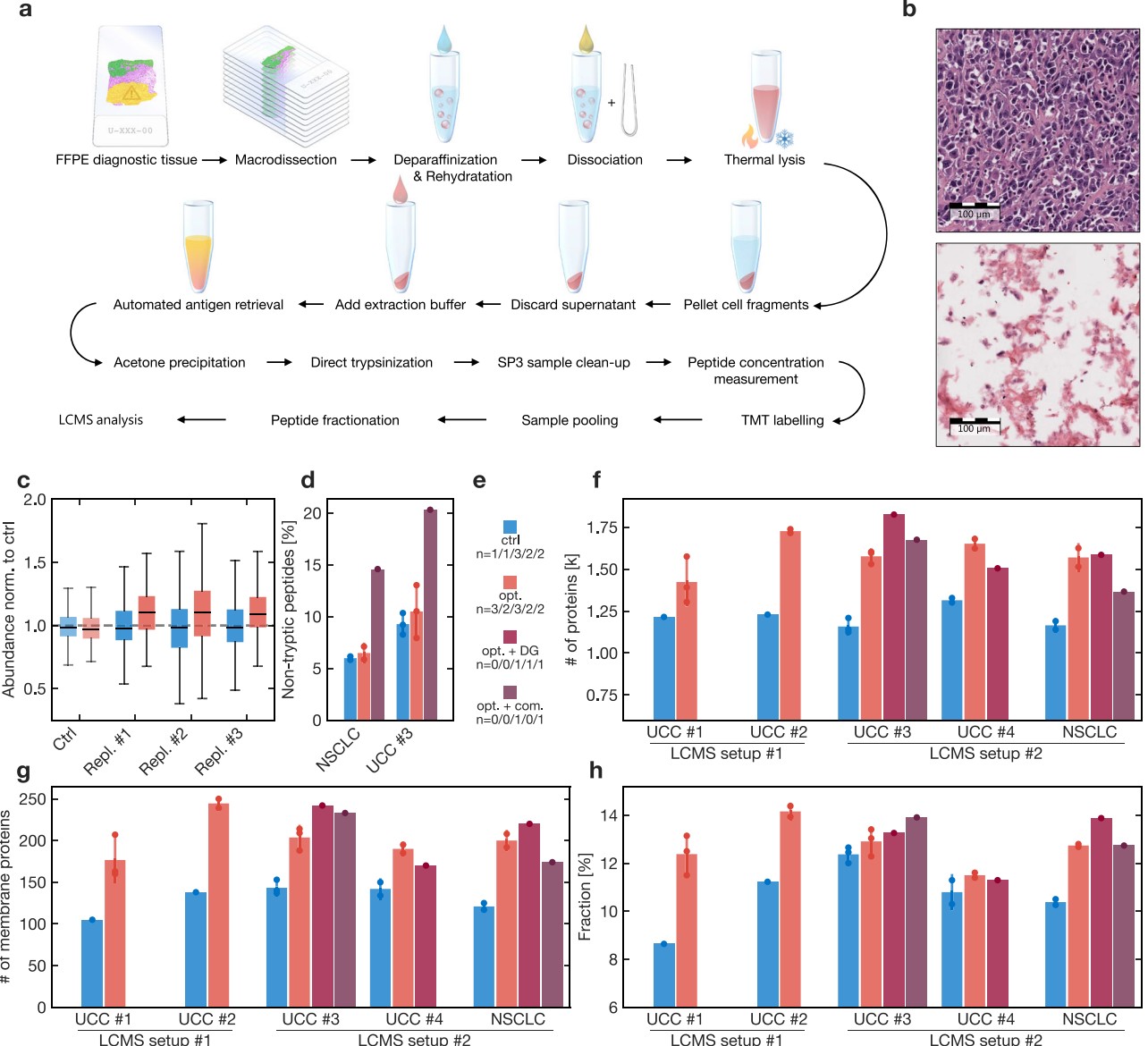

**Fig. 1 | Optimization of sample preparation by surface maximization and reduction of dimensionality (SMRD). a** Visualization of the pre-analytical sample preparation workflow and protein extraction; FFPE: formalin-fixed paraffin-embedded; LC-MS/MS: liquid chromatography-coupled tandem mass spectrometry; TMT: tandem mass tags. **b** Exemplary micrographs of input tissue (upper) and the resulting fragments (lower image) from the experiments in subplots **d**–**h**. **c** TMT-LC-MS/MS analysis of the standard protocol (Ctrl, lighter colors) and three technical replicates of the optimized protocol; protein intensities were normalized to a second control; relative abundances of membrane (red) and non-membrane proteins (blue) are shown as boxplots with median and inter-quartile range (IQR), whiskers extend to 1.5 IQR of lower and upper quartile respectively. **d** Fraction of non-tryptic peptides when using either a highly specific collagenase (red) or a commercial enzyme (mix) (dark red) for single-cell isolation compared to omission of any enzymatic matrix digestion (blue); bars are means, whiskers are standard deviations, dots are single data points (applies to all subplots **d**–**h**); NSCLC $n = 2/2/1$ technically independent samples; UCC $n = 3/3/1$. **e** Legend and number of replicates (by sample type) for subplots **f**–**h**; UCC#1/#2/#3/#4/NSCLC; Opt.: optimized; DG: deglycosylated; com.: commercial enzyme (mix); **f** Number of proteins identified in a single MS run without fractionation and without matching spectra between runs; please see methods for the different technical setups; **g** Number of membrane proteins; **h** Fraction of membrane proteins. Source data are provided as a Source Data file.

## Robust identification of proteomic clusters

On average, 5270 proteins were identified in each TMT set, with 9542 distinct and 2499 shared proteins across all 29 TMT sets (Fig. 2b). Principal component analysis of all 434 samples showed no relevant TMT set bias/co-clustering and indicated the separation of healthy and tumor samples (Fig. 3a). The common proteins of the 242 tumor samples were used for unsupervised consensus clustering with non-negative matrix factorization (NMF). The clustering performance peaked at k = 2 and k = 5 clusters, with k = 5 showing a highly distinct local optimum (Fig. 3b). The performance across multiple stochastic NMF runs (consensus) was robust (dispersion >0.8; Fig. 3b, c). Tumor

purity in the samples was comparable between all clusters (Supplementary Fig. 3), and survival stratification by cluster was independent of tumor purity, the order of pathological review and extraction, and the resulting peptide concentrations (Fig. 3).

## Proteomic clusters are independent prognostic factors

The five clusters mirrored the tumor biological spectrum of UC, e.g. separating early stage pTa from MIBC samples (Fig. 3d). Comparable to the survival stratification by pTNM stages, our 'Proteomic Analysis of the Urothelial cancer LAndscape' (PAULA) clusters demonstrated relevant stratification of overall survival (Fig. 3E; p < 0.0001; pairwise

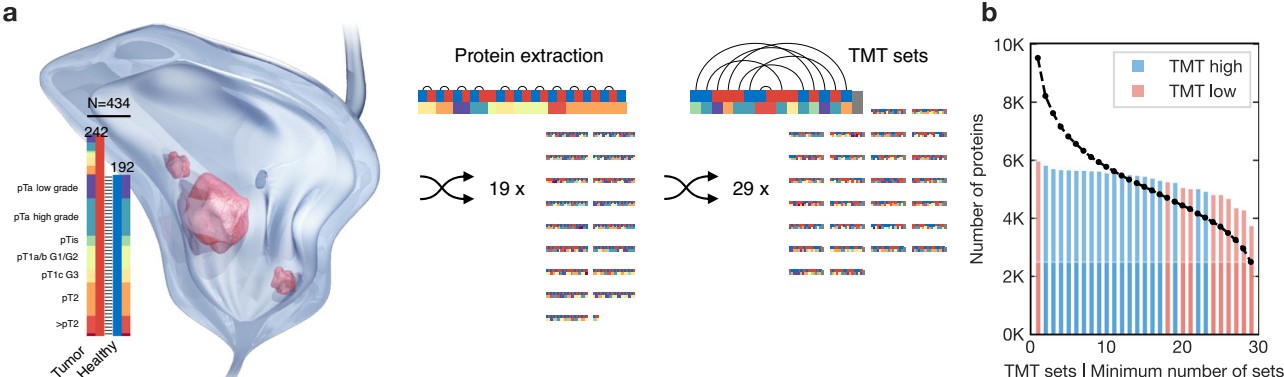

**Fig. 2 | Proteomic cohort characteristics and sample workflow. a** Schematic overview of the different steps of randomization; samples were randomized twice, at the stage of the protein extraction (keeping pairs and samples from the same patient together) and before tag labelling (again keeping pairs and patients together but randomizing across the tag channels); red and blue indicate tumor and healthy samples, respectively, spectral colors indicate pathological stages, arcs connect sample pairs; grey is the 134 N pooled reference standard; TMT: tandem mass tags; **b** Number of identified proteins across the different TMT sets of high and low peptide concentrations; dashed line shows the number of commonly identified proteins across the number of TMT sets. Source data are provided as a Source Data file.

comparisons in Supplementary Data File 7; without metachronous samples in Supplementary Fig. 4), with one low-risk cluster (PAULA I; 33% (n = 81) of all tumor samples; median survival 103 months, 95% confidence interval [81–120]), three intermediate-risk clusters (PAULA IIa/IIb/IIc; 16/11/18% (n = 38/26/44); 71 [37–inf]/58 [29–115]/72 [25–83] months), and one high-risk cluster (PAULA III; 22% (n = 53); 27 [16–49] months). However, PAULA clusters contained additional prognostic information, mirrored by their survival stratification of NMIBC samples (Fig. 3f) or pT1 G3 samples (Supplementary Fig. 5). Due to the lower sample sizes for non-III clusters, overall survival stratification was less clear in the MIBC subset, but external validation in the TCGA MIBC cohort (please see below) demonstrated similar prognostic stratification. Stratification of progression-free survival (PFS; Fig. 3g) revealed higher risk for PAULA III and IIa samples, also in MIBC (Supplementary Fig. 6).

Apart from age (hazard ratio (HR) 1.05, 95% confidence interval [1.02 − 1.07]), PAULA clusters and pTNM stages were the only significant risk factors in a multivariate Cox proportional hazards regression of overall survival (Fig. 3k, l). The HRs of the independent PAULA (1.19 [1.03 − 1.38]) and pTNM groups (1.22 [1.09 − 1.37]) were comparable (when sorted groups were treated as ordinal variables). Technical variables, such as TMT set, extraction order, peptide concentration, and specimen type, had no significant effect. PAULA III samples were also found in the NMIBC samples, with independent stratification of overall survival in these samples too (Supplementary Fig. 8).

We further integrated the information from the pTNM stages (pooling ≥pT2 due to a lack of overall survival stratification and difficult histological differentiation) and PAULA clusters (Fig. 3h) using a combined score $S$, with $S_{PAULA}$ being the score for the PAULA clusters sorted by overall survival (ranging from 1 to 5) and $S_{pTNM}$ being the pTNM stage as defined above (ranging from 1 to 6):

$$S = \frac{1}{5} S_{PAULA} \frac{1}{6} S_{pTNM} \qquad (1)$$

This increased the HR to 4.16 (2.49–6.95), while the Akaike information criterion decreased from 1115/1112 to 1108.

In patients with multiple metachronous samples (Fig. 3i), medium- and high-risk PAULA clusters preceded histopathological progression to MIBC in most cases. Similarly, PFS was shorter in PAULA IIa and III patients compared to PAULA I (difference of two-year restricted mean survival 3.6 and 3.9 months respectively; p < 0.005, respectively),

which was also observed in MIBC samples (two-year restricted mean survival for PAULA IIa and III 18.0 and 19.4 months vs. PAULA IIb and IIc 23.0 and 24.0 months). Supplementary Fig. 5–7 contain all survival stratifications. Supplementary Fig. 9 visualizes the stability of the proteomic subtypes in multiple metachronous samples from the same patients, where only a single 'downstaging' after the occurrence of a high-risk subtype was observed. Similarly, low-risk tumors rarely transformed into high-risk subtypes later.

## Proteomic clusters involve different tumor biological pathways

We evaluated the main oncogenic and tumor suppressor pathways using PROGENy (Fig. 3j)[34]. In direct comparison of tumor and healthy tissues, NFkB and VEGF were activated, whereas PI3K, TGF beta, and TNF alpha were inactivated. A subgroup analysis of the pTNM and PAULA subsets revealed further alterations: EGFR pathway activity increased with tumor stage (mirrored by EGFR expression, Supplementary Fig. 10a). PAULA I showed the strongest NFkB and VEGF activity, similar to early stage UC. PAULA IIb demonstrated prominently increased TGF beta activity, against the overall tumor trend. Most importantly, MAPK pathway activity was increased in PAULA III samples, whereas the pTNM subgroups showed increased activity only in the smaller pT1c G3 group. TNF alpha activity was retained in pTis and PAULA IIc samples, the latter entailing 10 (71%) of all pTis samples (but further 34 samples from different subgroups as well). Only the most advanced pN1 tumor samples exhibited relevant hypoxia activation.

## Tumor specificity is highly subset-dependent

One of the main objectives of our study was to meet the need for robust quantitative data on the specificity of potential descriptive protein targets for ADCs and other targeting strategies. The dataset can be accessed directly at www.cancerproteins.org. In pooled differential expression analyses between healthy and tumor samples (Fig. 4a) more proteins were downregulated, while several proliferation-related proteins (e.g., POL2RK, MEN1, MNAT1, TAF9, or SDHC) and biological processes (Supplementary Fig. 11) were upregulated. To fully exploit the value of our paired data, we evaluated significant over- and underexpression sample-wise by comparing each tumor sample to its individualized null distribution (Fig. 4b, see methods section for details). We then counted how often a protein was significantly (p < 0.05) over- or underexpressed within a sample subset, and report these as over- and underexpression rates for an individual protein.

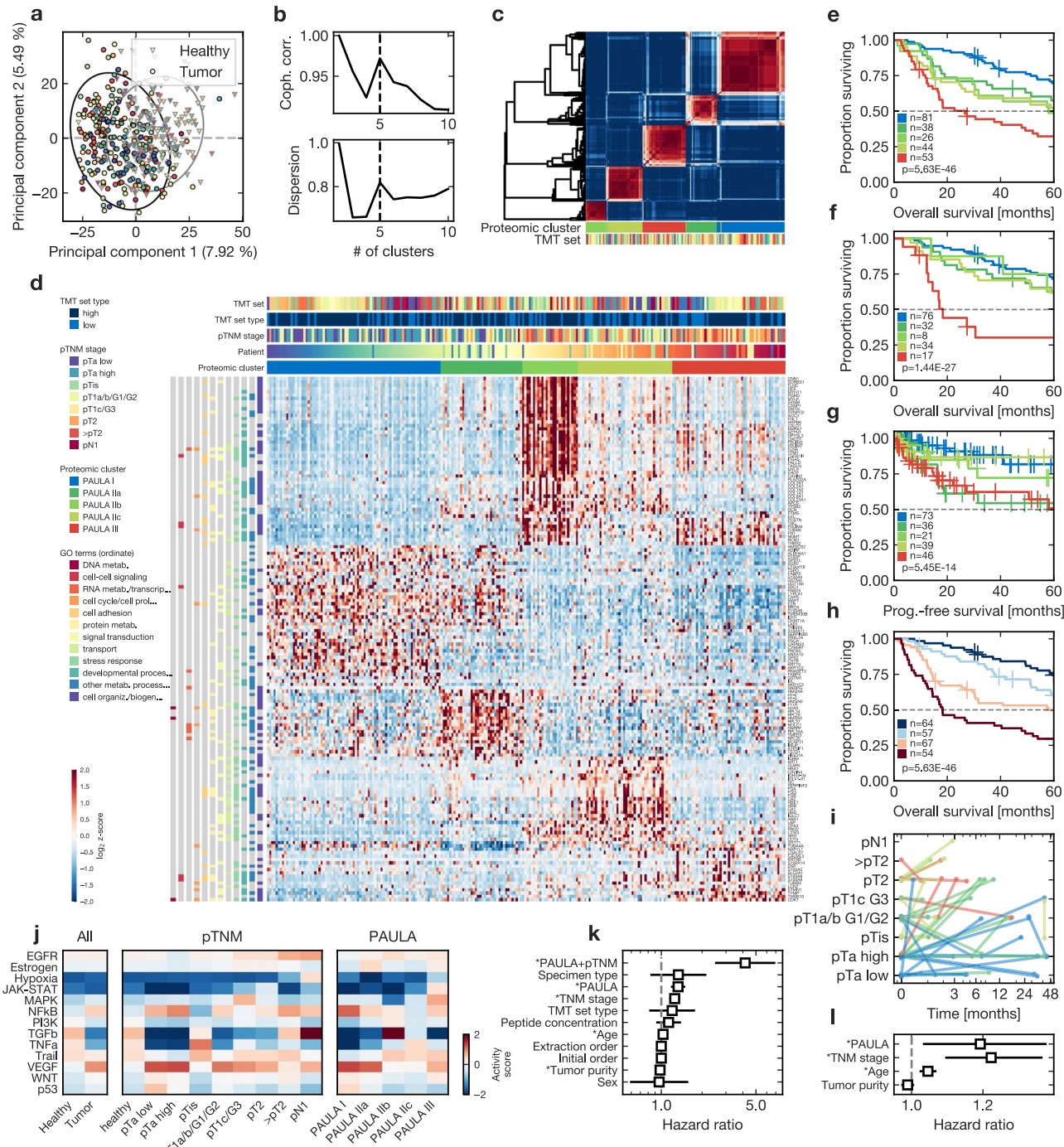

**Fig. 3 | Proteomic clusters map the urothelial bladder cancer landscape.**
**a** Principal component analysis of all samples; Tandem mass tag (TMT) sets in different colors; circles are the 95% confidence intervals; **b** Cophenetic correlation (upper panel) and dispersion (lower panel) as metrics of cluster rank selection; **c** Cluster stability across multiple clustering runs; samples on both axes; red: perfect co-clustering (in every stochastic run); blue: no co-clustering; **d** Abundance heatmap of all 242 tumor samples and the 25% most cluster-relevant proteins (n = 156) with clinical features and associated gene ontology biological processes; **e**–**h** Kaplan-Meier survival plots with stratification by proteomic clusters; p-values

are from log-rank tests; colors as in **d**. All stages (**e**), NMIBC (**f**), progression-free survival (**g**), and in combination with histopathological groups (**h**); **i** Histopathological groups for patients with multiple metachronous samples over time, corresponding proteomic clusters as colors; **j** PROGENy pathway analysis with activated (red) or inactivated (blue) oncogenic pathways; **k**, **l** Univariate (**k**) and multivariate (**l**) Cox proportional-hazards survival regression of all n = 242 biologically independent tumor samples; asterisk: p ≤ 0.05; squares are hazard ratios; bars are 95%-interval. Source data including the exact p-values for **k**, **l** are provided as a Source Data file.

Figure 4c visualizes these subset-specific expression rates of the 75 proteins with the highest general overexpression. No single protein exhibited general overexpression across all pTNM stages or PAULA subgroups. PAULA clusters demonstrated better separation with fewer groups (measured by the variance of the subgroup fractions; Fig. 4d).

By nature of our statistical approach, the overexpressed proteins showed high mean FCs with at least two up to eight-fold differences (Fig. 4e).

The nuclear protein S100P showed the highest general overexpression rate (26%) and a continuous stage-dependent decrease

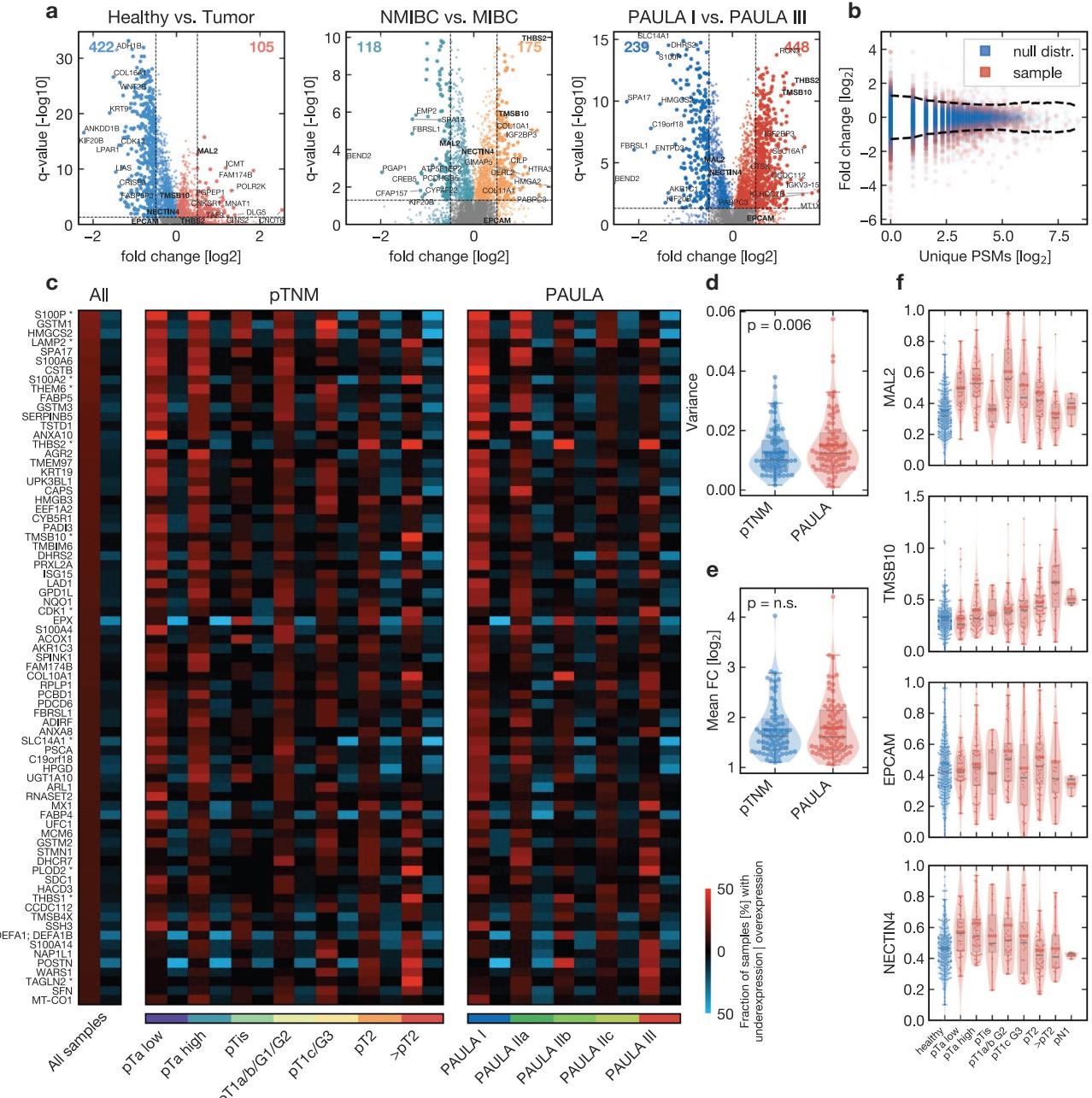

**Fig. 4 | Differential expression and tumor specificity profiles of individual protein markers. a** Pooled differential expression of all proteins with at least 20% coverage across the indicated subset samples (from left to right 6499, 6347 and 6343 proteins); Benjamini-Hochberg corrected Mann-Whitney-U q-values; dashed horizontal line is q = 0.05, vertical lines are absolute $\log_2$ fold change (FC) = 0.5; direction of differential expression as indicated by each subplot title; top 10 proteins by FC and q-value are annotated; selected proteins referenced in the text in bold font; **b** Exemplary plot of an individualized null distribution (blue) and the corresponding tumor profile (red); PSM: peptide spectrum match; the null distribution was obtained through pair-wise comparisons of the corresponding healthy mucosa with the other healthy mucosae of the same TMT set; the tumor profile was then compared to a PSM-filtered subset of this distribution; the dashed line shows the quantiles/significance thresholds (p < 0.05) for the tumor profile (please see methods for details); these profiles were calculated for all 192 paired samples

and were used to count the expression rates in subplot **c**. **c** The top 75 proteins with the highest tumor-specific overexpression across all samples and their expression rates across stages and subtypes (please see text for details; asterisk: referenced in the text); red/blue are the rates of significant over- and underexpression, respectively; **d** Variances of the group-wise overexpression rates for each protein from **c** with the samples being grouped by histopathological stages (pTNM) or proteomic (PAULA) subtypes; n = 75 proteins; violin plot and boxplot respectively, whiskers show 95%-interval, box is interquartile range, horizontal line is median, horizontal bar is mean (also in subplots **e**, **f**); p-value from two-sided Mann-Whitney-U (also in subplot **e**). **e** Mean fold change (FC) in analogy to subplot **d**; n = 75 proteins; **f** Exemplary expression profiles of the high-ranking proteins MAL2 and TMSB10 as well as the antibody-drug conjugate targets EpCAM and Nectin-4; n = 434/434/419/389 biologically independent samples, respectively. Source data are provided as a Source Data file.

with increased underexpression rates. Among further proteins of the S100 family, S100A2 showed a more generalized overexpression pattern, also in higher stages, with 13 to 42% across pTNM stages (except pTis).

Thymosin beta 10 (TMSB10; Fig. 4f), also found among the upregulated proteins from the standard differential expression analysis (Fig. 4a), was overexpressed in 52% of the >pT2 samples and in 15–30% of PAULA I, IIa, IIb, and III samples.

Further proteins with high subset-specific overexpression rates included the angiogenic protein thymidine phosphorylase (TYMP; 41% in PAULA III), the oncogenic protein cyclin-dependent kinase 1 (CDK1), and the metastasis-associated proteins Transgelin-2 (TAGLN2; 48% in >pT2) and PLOD2 (41% in PAULA III).

The transmembrane protein MAL2 was exclusively identified by standard differential expression analysis and showed early to medium stage upregulation (Fig. 4f). In contrast, the cell-surface lysosome-associated membrane glycoprotein 2 (LAMP2) showed high overexpression rates across all stages (21–32% in all PAULA subsets except IIa) and low underexpression rates in these subsets (< 3%; 25% in PAULA IIa). Other exemplary membrane proteins with high overexpression rates in our sample-wise analyses were THEM6, the sodium/hydrogen exchanger 1 (SLC9A1), the urea transporter 1 (SLC14A1), and Nectin-4, the latter showing relevant overexpression up to pT1 (overexpression 9–31%, pTis excluded because of low sample size; N.B.: underexpression 13% for pT1c G3; Fig. 4f and Supplementary Fig. 12). The secreted thrombospondins 1 and 2 (THBS1 and 2) showed advanced-stage overexpression (PAULA III 27% and 46%, respectively). EPCAM showed neither relevant sample-wise overexpression (overexpression: 0–12%; underexpression: 4–25%) nor pooled differential expression (Fig. 4f).

### Technical validation

To validate both SMRD sample processing and TMT-based quantification, we selected the large membrane protein integrin beta 1 (ITGB1) as an exemplary benchmark protein. We extracted proteins from a randomly selected subset of n = 67 samples, performed standardized immunoblots (IB; Fig. 5a, b) and measured the densitometric intensities of ITGB1 and the loading control cofilin-1 (CFL1). The normalized ITGB1 intensities showed a moderate positive correlation with the mass spectrometric abundances (Fig. 5b).

### Proteogenomic context with the TCGA dataset

To contextualize our PAULA clusters and validate their prognostic value, we used publicly available data from the TCGA cohort with the mRNA profiles of n = 408 MIBC samples[9]. To classify these transcriptomic samples within our proteomic PAULA scheme, we used published data on the correlations between proteins and their mRNAs across different tissue types[35]. These different biological states and cellular differentiations could be expected to represent maximum levels of biological regulation. Correlations from these samples were then likely to hold in the extrapolation to our data, too. We filtered for proteins with high correlation and transformed the datasets for conservative alignment of the two data distributions (Fig. 5c). The TCGA samples could be re-classified for 287 out of 408 samples with a clear prediction of ≥50% cluster probability/matching score.

We observed a general correlation of luminal (papillary) and basal (squamous) subtypes with PAULA I and III samples, respectively, whereas PAULA II subtypes were more heterogeneously associated (Fig. 5d, Supplementary Fig. 13[9,10,12,14,18,36]). Interestingly, almost all FGFR3 mutations (which are a hallmark of luminal tumors) were found in PAULA I samples (FDR-corrected vs. IIa/IIb/IIc/III p = 0.08/0.08/0.009/0.003), whereas they were found to be more evenly distributed across luminal papillary, luminal, and neuronal subtypes in the TCGA data (similar frequencies were found in basal squamous and PAULA III samples). TP53 mutations, in turn, were enriched in both PAULA III and PAULA IIa samples, which also showed subtle enrichment of hypomethylation clusters 4 and 1, respectively. Copy number variations, such as amplifications in PPARG, FGFR1, and NECTIN4, were more frequently observed in PAULA IIa samples, while neither CDKN2A (p16; as proposed marker for genomically unstable subtypes) gene alterations nor protein abundance (Supplementary Fig. 10b) showed an association. Interestingly, the higher NECTIN4 amplification rates in PAULA IIa

samples were mirrored by the higher protein abundances in PAULA IIa samples in our cohort (Supplementary Fig. 12). In our progression-free survival data (Fig. 3g), this molecularly distinct position of PAULA IIa was mirrored by diverging and reduced survival, similar to PAULA III samples. In general, the re-classified TCGA samples showed a survival stratification similar to our proteomic cohort (Fig. 5e).

Concerning further protein markers for the molecular (TCGA) subtypes, the luminal markers SRC, RAB25, E-cadherin, and ERBB2[13] demonstrated only mild overexpression in PAULA I samples (up to 13%), while basal-associated EGFR was overexpressed in 22% of PAULA III samples (2% in PAULA I). Of the consensus protein marker panel[37], the negative basal-squamous markers GATA3 and FOXA1 were identified in our dataset and were lower in PAULA III than in PAULA I samples (Supplementary Fig. 10c, d). In light of the use of these biomarkers to subtype samples by IHC, the distributions of the respective protein expression values overlapped considerably and did not stratify overall survival, which could have been expected. p16 (CDKN2A) was overexpressed in PAULA IIb, IIc, and III (Supplementary Fig. 10b).

### Immunohistochemistry reproduces PAULA clusters but lacks accuracy

Next, we investigated whether PAULA samples could be classified using readily available immunohistochemistry (IHC) instead of proteomic LC-MS/MS analyses. In analogy to the transcriptomic classifier, we transformed our data to a three-tiered system, paralleling the real-world achievable diagnostic IHC levels (absent, medium, and strong expression). Based on a comprehensive list of the most common diagnostic antibodies, we performed recursive in silico selection of the most frequent combination of features for maximum classification performance (Fig. 5f). We then evaluated the top seven proteins by IHC on a random subset of n = 54 samples. All antibodies were of validated diagnostic grade and all assays run and evaluated under diagnostic conditions. Collagen IV, osteonectin, coagulation factor XIII, and caldesmon were either technically not quantifiable or showed insufficient correlation with LC-MS/MS abundances (Fig. 5g). Tryptase (TPSB2) and CD163, however, could be quantified as discrete counts of qualitative IHC reactions (Fig. 5h), staining infiltrating immune cells in a binary fashion, rather than creating different staining intensities. Using this readout, the two IHC assays showed a positive quantitative correlation, while CD99 did not (expressed both on infiltrating leukocytes but also diffusely on urothelial cells[38,39]; Fig. 5g). Classifier performance for PAULA I, II (combining IIa, IIb, and IIc) or III based on tryptase and CD163 was within the range expected in silico (Fig. 5f, lower panel), with medium accuracy for PAULA I (81%) and II (76%), but low accuracy for PAULA III (18%; Fig. 5i). This was also in line with lower immune cell infiltration in re-classified PAULA I samples from the TCGA cohort (Fig. 5d) but comparable infiltrate in the other PAULA samples.

### In silico drug prediction suggests PAULA-specific compounds

To further evaluate the functional and biological relevance of the proteomic subtypes, we submitted our sample-wise over- and underexpression profiles to CMap[40], after filtering for high mRNA/protein correlation (n = 180; as described above). CMap compares qualitative lists of up- and downregulated transcripts to a library of perturbation profiles of drugs, knockouts, and proteins across several cell lines. Inversely correlated profiles suggest therapeutic efficacy. Counting occurrences across the PAULA subsets, Fig. 6a, b display the most frequently predicted pathways and drugs. High frequencies were observed, in part exceeding 50% of the subset samples. Most pathways were proliferation-related, including histone deacetylase (HDAC), mTOR, CDK, tubulin, and PI3K inhibition. The corresponding drug predictions included experimental inhibitors and well-known drugs (Fig. 6b). Flubendazole and vincristine were predicted for PAULA I and

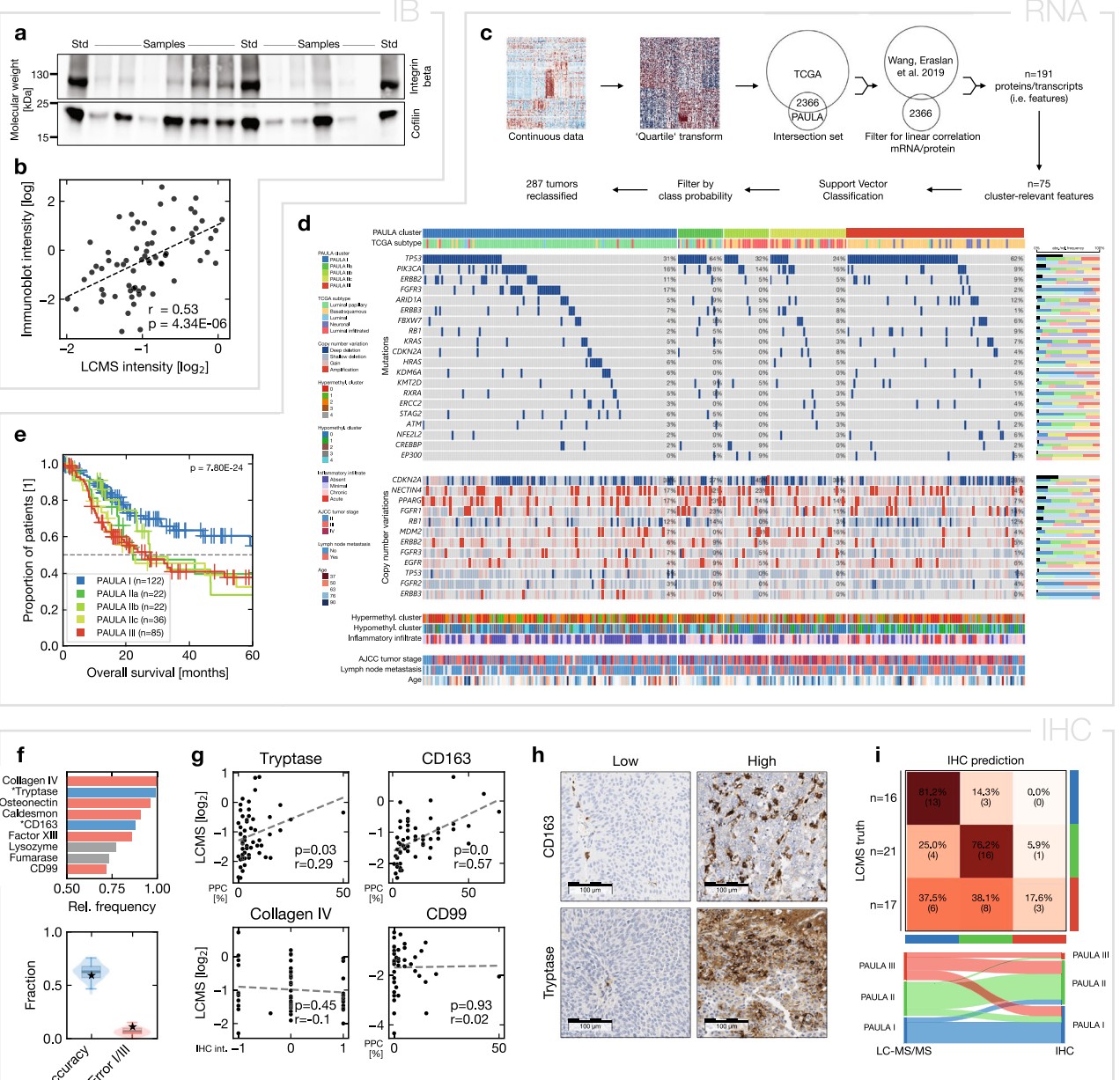

**Fig. 5 | Technical validation, transcriptomic and proteogenomic contextualization and data-driven immunohistochemical panel classifier.**
**a** Exemplary immunoblot (11 samples out of the n = 54 in the technical validation sub cohort across five immunoblots) of ITGB1 (Integrin beta) and CFL1 (Cofilin-1); Std: standard; **b** Correlation of immunoblot and LC-MS/MS abundances of ITGB1; Spearman's r; **c** Schematic overview of the classifier used for re-classification of the TCGA samples based on their transcriptomic data; **d** Mutational, structural variant, and epigenetic information from the reclassified TCGA dataset; top panel: customized OncoPrint of the 20 most frequent mutations; middle panel: corresponding relevant copy number variations; lower panels: further epigenetic and clinical data; small insets on the right visualize (per gene) the absolute frequency of mutations/amplifications/deep deletions (upper bar) and the relative frequencies per PAULA cluster (middle bar) or TCGA subtype (lower bar); **e** Kaplan-Meier overall survival plot of the validation dataset with all samples that could be reclassified with ≥50% cluster probability; p-value from log-rank test; **f** Upper panel: the top n = 9 proteins from the feature selection for the data-driven immunohistochemical (IHC) validation; red: insufficient correlation with LC-MS/MS; blue: final selection; grey: not evaluated; lower panel: in silico estimates (violinplot; boxplot: whiskers show 95%-interval, box is interquartile range, horizontal line is median, horizontal grey bar is mean) and observed values (asterisk) of accuracy and cluster I/III prediction error; **g** IHC and LC-MS/MS correlations of four exemplary proteins from the seven proteins investigated in subplot **f**; PPC: percentage of positive cells; **h** Exemplary micrographs, 25× objective; correlation coefficients r and p-values from Spearman's rank correlation; **i** confusion matrix and Sankey plot of the IHC cluster predictions. Source data are provided as a Source Data file.

IIa in up to one third of these samples. Conversely, the mTOR inhibitors NVP-BEZ235, AZD-8055, and KU-0063794 were found in 33 to 45% of PAULA III samples. PAULA clusters provided a more concise grouping of the samples, reaching similar or higher predictive fractions with fewer subsets than the pTNM groups (Supplementary Fig. 14a).

## Predictive correlation with PAULA clusters in vitro
To evaluate the top representative in silico drug predictions in vitro and to investigate whether PAULA clusters had functional/biological relevance for increased sensitivity towards these drugs, we selected 12 different UC cell lines as a model for functional patient samples across the UC spectrum. We evaluated the toxicity of seven different drugs,

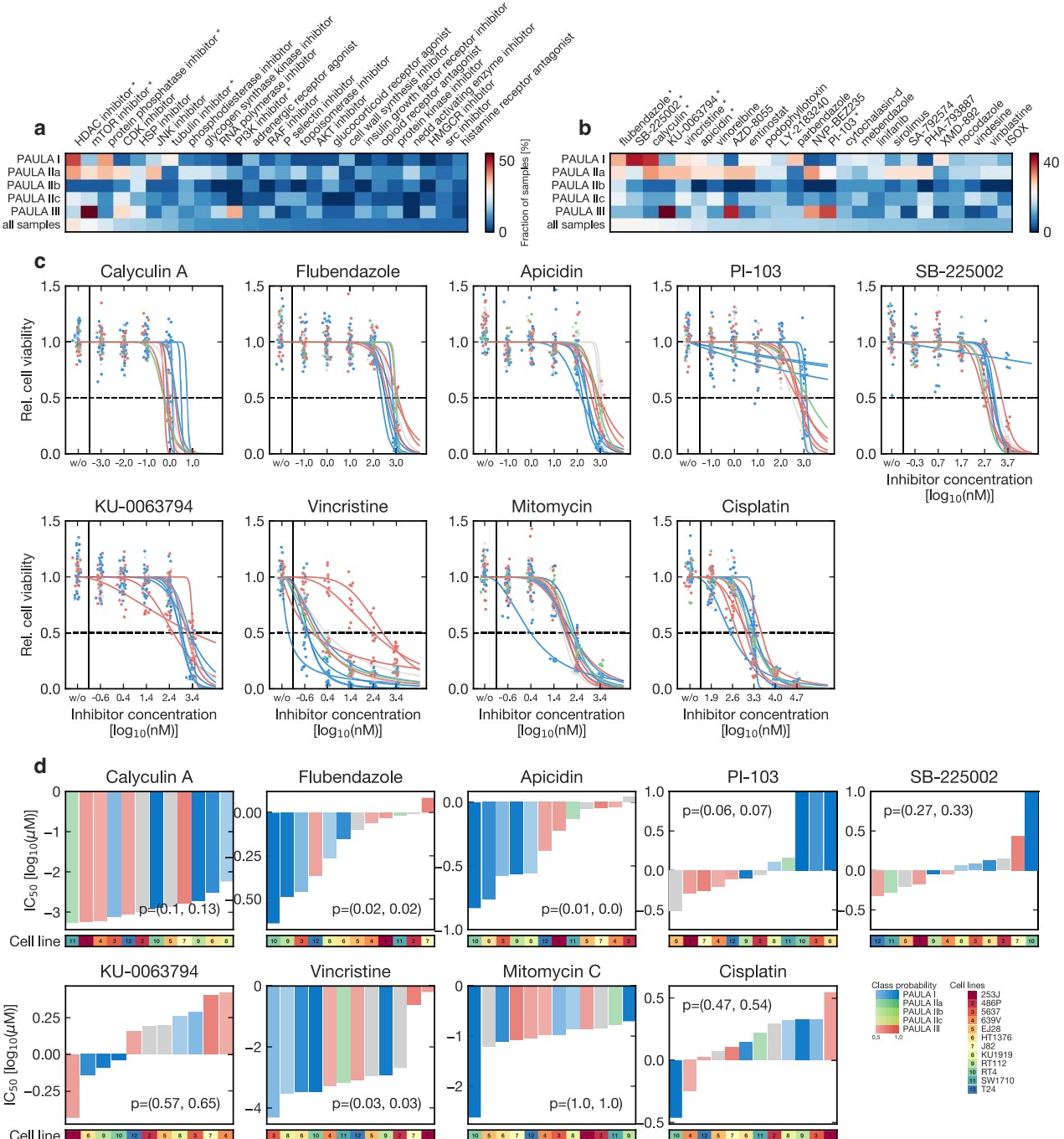

**Fig. 6 | Drug repurposing and predictive correlation with the proteomic subtypes.** Heatmaps of the frequencies of the in silico CMap predictions of **a** pathways and **b** drugs based on n = 180 qualitative profiles of up and downregulated proteins; asterisk: investigated in vitro. **c** In vitro dose-response curves of n = 12 urothelial cancer cell lines, which were selected as a functional UC panel and classified within the PAULA scheme; all cell lines were treated with exemplary compounds from the in silico predictions from A; dots are single data points, line is the fitted Hill function used to calculate half-maximal inhibitory concentrations ($IC_{50}$); colors as in **d**. **d** $IC_{50}$ values of the cell lines from C; in ascending order, bar base is 1 μM as indicator of sensitivity (bar downwards = more sensitive); color is predicted PAULA cluster based on RNA data, opacity is prediction confidence (cluster probability); p = (two-sided Kruskal-Wallis of sorted order, two-sided Mann-Whitney-U of quantitative values). Source data are provided as a Source Data file.

for which high cluster-specific sensitivity had been predicted, representing different molecular pathways (Fig. 6). We also included the direct DNA intercalator/crosslinker mitomycin C and cisplatin as the current standard therapeutics. We used further publicly available drug screening data for a total of 29 UC cell lines. All cell lines were classified within the PAULA scheme based on transcriptomic data, as described above.

Of the seven predictions, four could be confirmed (Fig. 6c, d). For PAULA I and IIa cell lines the tubulin inhibitors flubendazole and vincristine as well as the HDAC inhibitor apicidin showed significant relative and absolute $IC_{50}$ reductions compared to PAULA III samples, indicating increased sensitivity. Conversely, the combined mTOR/PI3K inhibitor PI-103 showed significantly lower $IC_{50}$ values for PAULA III samples. The mTOR inhibitor KU-0063794 generally showed high $IC_{50}$

values close to or in the micromolar range, hinting at low general sensitivity, whereas the most sensitive cell line belonged to the PAULA III group, as predicted. Further drug-screening data (Supplementary Fig. 14b) demonstrated general sensitivity towards the mTOR inhibitor AZD8055 and especially the combined mTOR/PI3K inhibitor NVP-BEZ235. The protein phosphatase inhibitor calyculin A showed very high general sensitivity across all samples ($IC_{50}$ around 1 nM), with a trend towards increased sensitivity in PAULA III samples. Predictions for SB-225002, a chemokine receptor antagonist with low general sensitivity, could not be confirmed. Both non-specific toxins, mitomycin C and cisplatin, did not exhibit subtype specificity.

## Discussion

Prognostic and predictive stratification of patients with bladder UC is clinically challenging, yet highly relevant[8]. Proteomics analyzes cellular effectors and offers direct quantitation of targets[41], which is vital information for tumor-specific targeted therapies such as ADCs. We advance proteomic analyses towards clinical LC-MS/MS applications and report the prognostic and predictive PAULA subtypes. We provide comprehensive information about the tumor specificity of a large part of the proteome and all stages of UC.

Sample degradation, endogenous necrosis, tissue composition, fixation duration, and other sample-handling factors impact negatively on protein quality[42,43]. These parameters are difficult to standardize in real-world clinical and diagnostic routine and are unknown for archival tissue. We propose SMRD, an optimized protocol for pre-analytical sample processing to reduce these effects, which are especially relevant in the frequently degraded urinary bladder specimens. We tested SMRD in three different LC-MS/MS setups, with label-free and TMT label-based quantitation. While the unoptimized controls were comparable to previously published proteomic analyses from single-run, single-replicate archival FFPE tissue[44–47], the SMRD-processed samples had considerably increased identification numbers and larger fractions of the highly relevant membrane proteins.

With the need for a comprehensive approach based on representative samples[48], we collected a quasi-consecutive, unbiased cohort covering both MIBC and NMIBC. In line with the continuous tumor-biological spectrum of UC[8], rather parallel than separate and in contrast to the MIBC and NMIBC dichotomy[49], we identified the proteomic PAULA clusters with survival stratification across the entire histopathological spectrum. While associated with histological UC stages, the PAULA clusters contained independent prognostic information. In the NMIBC subset, PAULA III samples were found in both pTa and pT1 samples and were independent predictors of reduced OS, whereas PAULA IIa was associated with reduced PFS and RFS. In MIBC, PAULA III and IIa were unfavorable for RFS and PFS. We used further, re-classified MIBC samples of the TCGA cohort as external validation of the prognostic value of the PAULA subtypes. Here, PAULA clusters also demonstrated relevant OS stratification. While PAULA I and III were associated with luminal(-papillary) and basal(-squamous) transcriptomic subtypes, respectively, PAULA II samples showed a distinct molecular pattern without clear association. With a pathway activation pattern similar to that of PAULA I, the survival data showed a similarity of PAULA IIa samples to PAULA III.

To date, only a few proteomic UC analyses have been conducted[50–52]. Latosinska, et al.[46] investigated the difference between invasive and non-invasive samples with a total of eleven samples. de Velasco, et al.[53] investigated a larger cohort of 58 MIBC samples. With fewer overall identifications, they identified two clusters, which showed non-significant stratification of distant relapse-free survival. They also constructed a proteomic signature based on focal adhesion proteins with independent prognostic significance. Stroggilos, et al.[44] performed LC-MS/MS analyses of a cohort of 19 MIBC and 98 NMIBC fresh-frozen (FF) samples considering only the NMIBC samples for cluster analysis. Based on fewer proteins, they identified three NMIBC

groups. The prognostic impact was estimated by preexisting risk scores without clinical follow-up data. Recently, Xu et al.[54] analyzed the proteome of 116 samples, in part with paired healthy tissue samples, in a label-free setup, matched between runs[55]. While the latter can be problematic in large series[56] they reported high identification numbers for their FFPE samples comparable to FF samples[57]. These were collected prospectively under study conditions, presumably with standardized short fixation times, explaining the difference to our and others' identification numbers with archival FFPE tissue[44–46,58]. Based on proteins expressed in 30% of all samples and with a dominance of MIBC samples (n = 71; 61%) they identified three proteomic clusters with significant survival stratification in univariate analysis (ibid.; Fig. 5b), very similar to the NMBIC/MIBC survival stratification (ibid.; Fig. 4a). Based on their Tbl. S1A, we additionally performed a multivariate analysis of these proteomic clusters (with pTNM groups in analogy to our subgroups), which did not reach significance as independent factors. In contrast to our approach, normalization was performed similar to total peptide normalization, and healthy and tumor samples were compared as groups.

We investigated if IHC with its widespread availability could be used as a surrogate method for proteomic subtyping. While we could reproduce the basic subtype composition of our cohort in a data-driven, real world-like approach, we generally observed insufficient correlation between IHC intensities and mass spectrometric abundances. Together with the known limitations of IHC-based protein quantitation[59], limiting its performance even in single-marker quantitation[60–62], reliable patient-specific subtype prediction using simple IHC appears questionable. As an example, Jackson, et al.[63] reported a small group of NMIBC samples with a basal-type expression profile and reduced GATA3 positivity in a three-marker IHC-based scoring system. Our quantitative LC-MS/MS data showed no indication of relevantly altered GATA3 levels (Supplementary Fig. 10d), which could be explained by IHC-specific confounders (such as more epitope-degraded samples with more advanced, frequently inflamed cancers). Vice versa, GATA3 expression by infiltrating lymphocytes could mask this effect in our bulk samples. As another example, overexpression of the proposed luminal/negative basal marker FOXA1[37] in luminal-like PAULA I samples was limited, and its expression levels did not stratify survival.

We used a conservative approach to individualize the expression differences between pairs of healthy and tumor tissue. In line with the broad tumor-biological spectrum of UC, overexpression rates (the fraction of samples with statistically significant overexpression) were highly subset-dependent. Even across samples of the same subset, rates were generally below 50%. Probably resulting from inter-individual tumor heterogeneity, these observations underline the necessity of personalized predictive testing. Relevant tumor heterogeneity has repeatedly been reported for UC and it is likely responsible for varying response rates[64]. All data, including protein-wise tumor specificity, is available at www.cancerproteins.org. Among the proteins with relevant overexpression, several were oncologically relevant. TMSB10 has been linked to progression and proliferation[65]. Elevated and prognostically relevant Cystatin B levels have similarly been described using 2D gel electrophoresis with LC-MS/MS in urine as well as IHC in tissue samples[51,66] (we could not confirm an increase with stage (ibid.), instead, overexpression rates decreased). TYMP, overexpressed in high-risk PAULA III samples, has demonstrated potential as an anti-angiogenic target, responding to inhibition with Tipiracil[67]. TAGLN2 has been associated with epithelial-to-mesenchymal transition and has previously been identified in paired urine and tissue samples[51,68,69]. It was found to respond to STAT3 inhibition[70], and like TAGLN2 itself, the JAK-STAT pathway was relevantly activated only in PAULA III samples. Thrombospondin-1 and -2 showed very high overexpression rates in invasive stages and PAULA IIb and III clusters (up to 50%). THBS2 has been identified as an adverse prognostic factor in UC[71]

and in colorectal cancer[72], and appears to increase immune cell infiltration (ibid.). THBS1 has been shown to activate the TGF beta pathway[73], which correlated with THBS1 abundance in our data, too. Several members of the S100 protein family demonstrated high overexpression rates, especially in NMIBC. Amongst these, S100P showed the highest overall rate of all proteins in our study, but lower than IHC suggested[74] and with high underexpression rates in MIBC. S100 proteins have been associated with an unfavorable prognosis and cisplatin resistance[75], increased proliferation[76], and altered PPP5C protein phosphatase activity[77]. Interestingly, PPP5C has been shown to be inhibited by Calyculin A[78], which was one of the most frequently predicted drugs in our data (please see below).

Concerning potential ADC targets, the membrane/lysosomal protein LAMP2 was one of the most promising candidates, showing relatively uniform overexpression across subgroups. On average, however, overexpression was found in only 22% of all samples. LAMP2 has been linked to different cancer relevant processes in different entities[79]. The plasma membrane protein MAL2 was identified in the pooled differential expression analysis across all samples. It modulates the immune response and appears to have complex effects in tumors[80]. It has been shown to correlate negatively with immune cell infiltration in breast cancer[81] and is associated with (mostly) worse prognosis across various cancer types[82] with limited data for UC. In light of the heterogenous response rates to immune checkpoint inhibitors in UC, high MAL2 expression could serve as a negative predictor of immune checkpoint blockade, while its inhibition could potentially restore efficacy. Its individualized overexpression rates, however, were lower (for instance PAULA I 22%, PAULA III 7%), again underlining the need for proteomic predictive testing.

The Nectin-4-targeted ADC Enfortumab Vedotin has recently been approved in combination with Pembolizumab as first-line treatment for advanced or metastatic MIBC. Further analyses have evaluated Nectin-4 expression in NMIBC with regard to an extended use. We observed varying medium overexpression in early stages, in line with its low tissue specificity as target molecule of the measles virus[83]. IHC data had suggested much higher overexpression in tumor tissue, whereas a decrease in invasive areas was already noted[28]. The former probably originates from different cohorts for the two sample types with a high probability of the healthy samples being biased by degradation (ibid.). Varying over- and underexpression rates with associated changes in clinical outcome have previously been indicated by IHC[84] and on transcriptome level, too[85]. Our data underlines the necessity of predictive testing for this biomarker (e.g., with adjacent normal tissue as internal standard). Interestingly, PAULA IIa samples demonstrated increased abundance of Nectin-4 and were associated with *NECTIN4* amplifications in the re-classified TCGA samples.

For EpCAM, the target of the ADC Oportuzumab Monatox, our data did not indicate relevant overexpression, in line with its ubiquitous epithelial expression[24,27].

Besides surgery, UC treatment depends heavily on the efficacy of chemo- and immunotherapy with limited response rates. We investigated whether our proteomic subtypes would correspond to specific predictions by in silico drug repurposing. Replicated CMap analyses yielded several cluster-specific pathways and compounds. We selected exemplary drugs for the treatment of a representative cell line panel, which we classified within the PAULA scheme, to investigate if the proteomic subtypes would correlate with increased sensitivity in vitro. This way predictions for four out of seven drugs were confirmed. Promising a priori due to its frequent occurrence amongst high-risk PAULA III samples, the mTOR inhibitor KU-0063794 did not exhibit sufficient sensitivity. Similarly, Mokou et al.[86,44] had identified mTOR inhibitors in NMIBC, but their validation also showed low sensitivity with effective doses in the micromolar range. Clinically, single-arm trials with everolimus lead to partial and complete remissions only in very few cases[87,88]. In contrast, the combined mTOR/PI3K/DNA-

dependent protein kinase inhibitor PI-103 demonstrated PAULA III-specific sensitivity. Further drug screen data (Supplementary Fig. 14b) showed similar results for the combined PI3K/mTOR inhibitor NVP-BEZ235 (Dactolisib; with a trend of PAULA III specificity). The efficacy of PI3K-AKT inhibition in UC has been indicated by Lv et al.[89]. Subset specific combined inhibition could thus offer another therapeutic strategy in advanced UC, especially as approved drugs are available for these separate pathways, too (e.g., everolimus + alpelisib). For PAULA I samples, the tubulin inhibitors flubendazole (approved as anthelminthic but with anti-neoplastic effects[90,91]) and vincristine showed increased sensitivity. Similarly, the HDAC inhibitor apicidin demonstrated high PAULA I specificity. Apicidin has previously been reported to inhibit proliferation of breast, endometrial and ovarian cancer cells[92,93], and HDACs have been linked to UC[94–96]. The protein phosphatase inhibitor calyculin A showed very high general sensitivity (around 1 nM), much higher than the current standard therapeutic mitomycin C. As a membrane-permeable fungal toxin, it enters cells easily and inhibits several protein phosphatases (also PPP5C; please see above). Calyculin A has demonstrated apoptosis-inducing antitumoral effects[97,98] but its toxicity extends to healthy tissue with evidence of non-specific intracellular calcium increase[99]. We report the first data in UC cell lines, but off-target toxicity still needs to be evaluated. Promisingly, tumor-toxic concentrations of calyculin A did not reduce neither cell proliferation nor viability in non-tumorous human umbilical vein endothelial cells[100].

As a limitation, contamination by stromal tissue and immune cells cannot be avoided, especially for infiltrative samples and despite thorough microdissection of all samples in our study. With increasing availability, semi-automated microdissection[101] could lead to further improvement in this area. Concerning an ideal proteomic sample preparation strategy, we observed that balancing deep sample extraction and sample loss is pivotal. Our proposed SMRD protocol could potentially further be refined with regard to the latter, e.g., by optimizing binding of the cell fragments to carboxylate paramagnetic beads upfront. Last, all in vitro experiments were performed with a cell line panel of limited size and the cell lines were classified indirectly. While significant predictive correlations were observed, independent validations are necessary with a higher number of replicates.

In summary, we report proteomic molecular subtypes as highly distinct features of UC with prognostic value independent of histopathological stages or the NMIBC/MIBC dichotomy. We further show predictive associations: Several drugs were identified and validated in vitro, which are hitherto not in use. In view of many forthcoming ADCs and other targeted therapies, we provide target specificity data (available at www.cancerproteins.org) and demonstrate that generalized overexpression cannot be assumed. Instead, individual predictive testing should be the rule not the exception. In contrast to transcriptomics and IHC, only proteomics can provide this essential information. To this end, our optimized SMRD protocol helps to advance clinical proteomics towards real-world diagnostics.

## Methods

All experiments were carried out and all data and tissue samples collected with a positive ethics vote of the Ethics Committe of the University of Lübeck and in compliance with state, federal, and European laws. The study design and conduct complied with all relevant regulations regarding the use of human study participants and was conducted in accordance to the criteria set by the Declaration of Helsinki. Authorization was granted by the Ethics Committe of the University of Lübeck (19-321).

### Study cohort

All tissue samples were collected retrospectively from a quasi-consecutive series across a five-year period. All samples were from

routine diagnostics and with the diagnostic procedure completed well before selection for the study.

## Sample preparation

20 μm thick sections were sequentially cut from FFPE tissue blocks and mounted on standard glass slides (ThermoFisher 15545650, Schwerte, Germany). One section from the middle of a series of up to ten sections was stained automatically with hematoxylin and eosin (H&E) on the Ventana HE 600 platform (Roche, Basel, Switzerland) and segmented as template by a pathologist. Depending on the resulting sample area, up to ten serial sections were manually dissected using the H&E template to yield a total sample area of ca. 1 cm². The tissue was transferred to 2 ml tubes with low protein adsorption (ThermoFisher 88379, Schwerte, Germany) and stored at −80 °C until processing.

## Custom dissociation device

For the mechanical tissue dissociation during the SMRD protocol (see below) we created a custom dissociation stir bar. We used 4301 V2A stainless steel wire (0.6 mm, Bauhaus 10198503, Mannheim, Germany), which proved to be chemically inert. Bend in a drop-like shape (Supplementary Fig. 1d) to fit in a 2 ml flat-bottom tube (see above), these bars created subtle mechanical shear stress without jamming when put in relative motion to the surrounding tube on a rotating mixer (at 1200 rpm; Eppendorf 5382000015, Hamburg, Germany).

## SMRD protocol

Samples were parallelly processed in randomized sets of up to 12 sample pairs (24 samples overall). Samples were deparaffinated twice by the addition of 500 μL xylol, incubation for 10 min at room temperature, brief centrifugation at $500 \times g$ for 2 min and careful xylol removal. Samples were then rehydrated by addition of 500 μL of 100/100/70/50/0% ethanol (v/v) respectively, incubation for 30 s with subsequent brief centrifugation with a tabletop centrifuge and careful removal of the solvent without letting the tissue dry. Samples were then resuspended and vortexed in 100 μL collagenase buffer C* (PBS 35% (v/v), Tris-HCl 100 mM, CaCl₂ 1 mM, Tween-20 0.05% (v/v), pH 7.40) and incubated at 90 °C for 60 min (for reduction of fixation effects prior to collagenase treatment). Collagenase in buffer C* (Merck C0773, Darmstadt, Germany) was added to a final activity of 250 U/ml. A custom-made stir bar was added to each sample tube and the tissue dissociated both mechanically and enzymatically for 120 min at 40 °C with 1200 rpm on a mixer (Eppendorf 5382000015, Hamburg, Germany). The stir bar was removed, and the tissue gently pipetted up and down (with a 1 ml tip) to complete dissociation. For cell lysis, the samples were slowly frozen at −20 °C (in a pre-chilled thermoblock) and then heated to 99 °C for 5 min. After cooling down on ice and brief centrifugation to spin down condensate, the samples were carefully resuspended and transferred to a 0.2 ml PCR tube including the remaining tissue (again with a 1 ml tip). Samples were then centrifuged at $5000 \times g$ and 4 °C and the resulting supernatant was discarded (to reduce the cytosolic fraction). The pellet was then resuspended in 25 μL extraction buffer S* (SDS 2% (w/v), Tris-base 200 mM, EDTA 1 mM, beta-mercaptoethanol 4.7% (v/v), proteinase and phosphatase inhibitor 0.94% (v/v; ThermoFisher 78840, Schwerte, Germany), pH 7.2). After incubation on ice for 5 min, the samples were vortexed, placed in a thermocycler, incubated at 4 °C for 5 minutes, at 90 °C for 30 minutes and then subjected to four cycles of 99 °C for 5 min followed by 60 °C for 10 min. The complete sample including any pellet if present was then transferred to a 1.5 ml low bind tube (ThermoFisher 90410, Schwerte, Germany) for detergent removal: 100 μL of ice-cold acetone were added and the samples incubated at −20 °C for 60 min. After centrifugation at $10,000 \times g$ for 10 min the supernatant was carefully discarded. The pellets were then resuspended in 5 μL buffer R (RapiGest 1% (w/v; Waters 186001861, Eschborn, Germany),

ammonium bicarbonate 25 mM) and 5 μL of dithiothreitol 50 mM (in ammonium bicarbonate 25 mM) was added. Samples were incubated at 37 °C and 950 rpm for 60 min. Then, 5 μL iodocetamide 100 mM (in ammonium bicarbonate 25 mM) was added and the samples were incubated at 37 °C and 950 rpm for 60 min again. The samples were then diluted with 45 μL of ammonium bicarbonate 25 mM and 10 μL of digestion mix were added, containing Trypsin and Lys-C at 1 μg/μL (Promega V5073, Madison, USA). Samples were then digested overnight (16 h) at 37 °C. The next day, digestion was stopped with 10 μL trifluoracetic acid 5% (v/v). After final incubation at 37 °C and 950 rpm for 60 min, the samples were centrifuged at $9000 \times g$ for 30 min and the supernatant stored at −80 °C.

Of note, further membrane protein enrichment strategies were tried but proved unsuccessful or not scalable to cohort analyses. Supplementary Note 1 reports these negative results.

## Cell dissociation staining

For the development of the SMRD protocol, samples were visualized by Cytospins™ (ThermoFisher 11630601 and 11610591, Schwerte, Germany) following the manufacturer's protocol using 5 μL of dissociated cell or cell fragment solution. The spins were H&E stained as described above.

## LC-MS/MS analyses and evaluation−setup #1

After lyophilization for 4 h, the samples were resuspended in 20 μL acetonitrile (ACN) 2% (v/v) including formic acid 0.5% (v/v). "Samples were then loaded onto a C18 column (15 cm, 3-μm Luna Phenomenex) in an Ultimate 3000 RSLCnano high-performance liquid chromatography system (HPLC; ThermoFisher) and injected online into a 5600+Triple ToF mass spectrometer (AB Sciex) in data-dependent mode (DDA) with selection of 30 precursor ions."[102] The raw data was evaluated (independently for each sample) using ProteinPilot (5.0% local FDR; version 5.0.2; AB Sciex, Darmstadt, Germany).

## LC-MS/MS analyses and evaluation−setup #2

After digestion, peptide cleanup was performed using an adjusted version of the SP3 protocol[103], In brief, 40 μg of Sera-Mag Magnetic carboxylate modified particles (1:1 mix of hydrophilic and hydrophobic; GE Healthcare) were added and mixed. ACN was added to a final concentration of 95% mixed for 10 min. Samples were placed on a magnetic rack, left for 1 min to settle and the supernatant was removed. After one wash with 100% ACN, peptides were eluted with 2% dimethyl sulfoxide (DMSO) and dried in a vacuum centrifuge.

Samples were reconstituted in 0.1% FA and transferred into a full recovery autosampler vial (Waters). Chromatographic separation was achieved on a Dionex Ultimate 3000 UPLC system (Thermo Fisher Scientific) with a two-buffer system (buffer A: 0.1% FA in water, buffer B: 0.1% FA in ACN). Attached to the UPLC was an Acclaim PepMap 100 C18 trap (100 μm × 2 cm, 100 Å pore size, 5 μm particle size, Thermo Fisher Scientific) for desalting and purification followed by a nanoEase M/Z peptide BEH130 C18 column (75 μm × 25 cm, 130 Å pore size, 1.7 μm particle size, Waters). Peptides were separated using a 60 min gradient with increasing ACN concentration from 2% to 30% ACN. The eluting peptides were analyzed on a Q Exactive hybrid quadrupole Orbitrap mass spectrometer (Thermo Fisher Scientific, Schwerte, Germany) in data-dependent acquisition (DDA).

The Q Exactive was used to analyze the top 12 most intense ions per precursor scan ($1 \times 10^6$ ions, 70,000 resolution, 120 ms max. fill time) and were analyzed by MS/MS with detection in the Orbitrap (HCD at 25% normalized collision energy, 17,500 resolution, $1 \times 10^5$ ions, 50 ms max. fill time) in a range of 400–1300 $m/z$. A dynamic precursor exclusion of 20 s was used.

Acquired DDA LC-MS/MS data were searched (independently for each sample) against the reviewed human protein database downloaded from Uniprot (release April 2020, 20,365 protein entries) using

the Sequest HT algorithm integrated in the Proteome Discoverer software v2.4 (Thermo Fisher Scientific, Schwerte, Germany). Mass tolerances for precursors was set to 10 ppm and 0.02 Da for fragments. Carbamidomethylation was set as a fixed modification for cysteine residues and the oxidation of methionine, deamidation on asparagine (only for the deglycosylated samples), pyro-glutamate formation at glutamine residues at the peptide N-terminus as well as acetylation of the protein N-terminus, methionine loss at the protein N-terminus and the acetylation after methionine loss at the protein N-terminus were allowed as variable modifications. Only peptides with a high confidence (false discovery rate <1% using a decoy data base approach) were accepted as identified. Peptide chromatographic peak areas were summed to protein abundances and used for quantitative analysis. Protein abundances were imported into Perseus software version 1.5.8 for statistical analysis.

### Tandem mass tag peptide labelling−cohort setup
In analogy to the SP3-like cleanup in setup #2 the peptide samples were cleaned on a C18 Hypersep plate with 40 μL bed volume (Thermo Fisher Scientific) and dried using a vacuum concentrator (Eppendorf). Peptides were dissolved in 50 μL MilliQ water and a 5 μL aliquot was taken to microBCA peptide concentration after 10-fold dilution. Depending on the total amount of peptides, either 5 or 10 μg samples in 70 μL of 50 mM triethylammonium bicarbonate (TEAB) were labeled with TMTpro reagents in random order adding 60 μg TMTpro-reagent in 30 μL anhydrous ACN and incubating at room temperature (RT) for 2 h. The labeling reaction was stopped by adding 11 μL of 5% hydroxylamine. The samples were incubated at RT for 15 min before combining them in one vial. To enable normalization of peptide abundances between the TMT sets and to remove the batch effect, a pooled sample was created comprising 50 biological samples, labeled with TMTpro-134N and added to each TMT set. For quality control, an aliquot of each TMT set was cleaned on StageTip C18 (Thermo Fisher Scientific) and analyzed by LC-MS/MS. The TMTpro-labeled tryptic peptides were dissolved in 50 μL of 20 mM ammonium hydroxide and were loaded onto an XBridge bridged ethyl hybrid C18 UPLC column (2.1 mm inner diameter × 250 mm, 2.5 μm particle size, Waters), and separated with a linear gradient of 5−60% 20 mM ammonium hydroxide in ACN (pH 10.0) over 48 min, at a flow rate of 200 μL/min. The chromatographic performance was monitored by sampling eluate with a UV detector (Ultimate 3000 UPLC, Thermo Fisher Scientific) scanning at 214 nm. Fractions were collected at 30 s intervals into a 96-well plate, and 8 fractions were combined into 12 samples of each TMTpro-labeled set before dried on a vacuum concentrator.

### LC-MS/MS analyses and evaluation−TMT
Peptides in TMT sets were reconstituted in solvent A and approximately, 1.8 μg of sample (4 out of 15 μL) injected on a 50 cm long EASY-Spray C18 column (Thermo Fisher Scientific) connected to an UltiMate 3000 nano-flow UPLC system (Thermo Fisher Scientific) using a 90 min long gradient: 4-26% of solvent B (98% ACN, 0.1% FA) in 90 min, 26-95% in 5 min, and 95% of solvent B for 5 min at a flow rate of 300 nL/min. Mass spectra were acquired on a Orbitrap Fusion Lumos tribrid mass spectrometer (Thermo Fisher Scientific) ranging from $m/z$ 375 to 1600 at a resolution of R = 120,000 (at $m/z$ 200) targeting $4 \times 10^5$ ions for maximum injection time of 50 ms, followed by data-dependent higher-energy collisional dissociation (HCD) fragmentations of precursor ions with a charge state 2+ to 6+, using 45 s dynamic exclusion. The tandem mass spectra of precursor ions were acquired with a resolution of $R = 50,000$, targeting $5 \times 10^4$ ions for maximum injection time of 86 ms, setting quadrupole isolation width to 0.7 Th and normalized collision energy to 35% with 2 s cycle time.

Acquired raw data files were analyzed using Proteome Discoverer v2.5 (Thermo Fisher Scientific) with Mascot Server v2.5.1 (Matrix Science Ltd., UK) search engine against the human protein database

(SwissProt) and a separate fasta file composed of 245 most common contaminant proteins (obtained from the MaxQuant software). The contaminant protein identifications were filtered out automatically in the data analysis workflow. A maximum of two missed cleavage sites were allowed for full tryptic digestion, while setting the precursor and the fragment ion mass tolerance to 10 ppm and 0.02 Da, respectively. Carbamidomethylation of cysteine was specified as a fixed modification, while TMTpro on lysine and N-termini, oxidation on methionine as well as deamidation of asparagine and glutamine were set as dynamic modifications. Initial search results were filtered with 5% FDR using Percolator node in Proteome Discoverer. Quantification was based on the TMT-reporter ion abundances.

### Standard protein extraction
Macrodissected tissue (cumulative 1 cm² à 20 μm) was scratched into 2 ml tubes with low protein adsorption (ThermoFisher 88379, Schwerte, Germany). Samples were incubated for 15 min in 2 ml xylol, repeated once, and washed with ddH₂O for 30 s. The tissue was then transferred to 0.2 ml PCR tubes and briefly spun down using a tabletop centrifuge. Residual water was removed and 60 μL extraction buffer were added (SDS 2% (w/v), Tris-base 200 mM, EDTA 1 mM, beta-mercatptoethanol 4.7% (v/v), proteinase and phosphatase inhibitor 0.94% (v/v; ThermoFisher 78840, Schwerte, Germany), pH 7.2). In analogy to SMRD, samples were vortexed after incubation on ice for 5 min, placed in a thermocycler, incubated at 4 °C for 5 min, at 90 °C for 90 min and then subjected to four cycles of 99 °C for 5 min followed by 60 °C for 10 min. The samples were vortexed again and debris removed by centrifugation at 10,000 × g for 15 min. The extracts were stored at −80 °C for a maximum of 8 weeks.

### Protein concentration determination
The commercial EZQ quantification kit (R33200, Invitrogen Thermo-Fisher, Waltham USA) was used as per the manufacturer's instructions. Ovalbumin standards were dissolved in extraction buffer as described above. Fluorescence was imaged on a Gel Doc XR+ imaging system (BioRad 1708195, Feldkirchen, Germany). Regression and calculation were performed in Microsoft Excel 16.64 (OS; Microsoft, Seattle, USA).

### Immunoblotting
Briefly, samples were prepared using a commercially available MOPS buffer system (NP0007, NP0004, NP0050, ThermoFisher, Schwerte, Germany) in 15-well 10% pre-cast Bis−Tris gels (minimum 10, optimum 30 μg per lane; ThermoFisher NP0303BOX) for 35 min with voltage and current limited to 200 V and 250 mA. Samples were then transferred to nitrocellulose membranes with 0.45 μm pore size (ThermoFisher LC2001) in transfer buffer (ThermoFisher NP00061) at 30 V and 250 mA for 60 min. Cut membranes were blocked in 20 ml PBS-T-M (PBS pH 7.4 (Merck P3813), Tween-20 0.5%, milk powder 5%) for 1 h. After washing twice (with 20 ml PBS-T) membranes were incubated overnight at 4 °C on a rolling shaker in 6 ml PBS-T-M including the primary antibody. The next day, membranes were washed four times (20 ml PBS-T; 3 min) and incubated with the secondary antibody for 2 h at room temperature (20 ml PBS-T-M). Washing was repeated and membranes were visualized on a densitometric imager (Amersham Imager 600, GE Healthcare 29083461, Freiburg, Germany) using ECL developing agents (GE Healthcare Life Sciences Europe RPN2106, Eindhoven, Netherlands) and quantified with ImageLab (6.0.1; BioRad, Feldkirchen, Germany). Both anti-ITGB1 (monoclonal rabbit IgG; Cell Signaling 9699 S, Leiden, Netherlands) and anti-CFL1 (monoclonal rabbit IgG; Cell Signaling Technology 5175 S, Danvers, USA) were used in 1:1000 dilution with the secondary antibody in 1:2500 (Thermo-Fisher 31460). Prerequisites for relative protein quantitation were accurately followed as described by ref. 104. Protein-specific densitometric intensities were first divided by the respective mean standard intensity before normalization with the loading control[105].

## Immunohistochemistry

Immunohistochemistry (IHC) was performed on the Ventana Benchmark Ultra platform. The antibodies were Caldesmon (Cell Marque E89, Rocklin, USA, ready-to-use (RTU)), CD99 (Agilent 12E7, Santa Clara, USA; RTU), CD163 (Cell Marque MRQ-26; RTU), collagen IV (Cell Marque CIV22; RTU), factor XIII (Cell Marque AC-1A1; RTU), tryptase (Cell Marque G3; RTU) and osteonectin (Leica 4A4, Wetzlar, Germany; 1:10 dilution). The stained slides were microscopically evaluated by a pathologist and quantified as stated in the results section under real-world conditions.

## Cell lines and cell culture

Cell lines were freshly obtained from Merck (HT1376; 87032402/RRID: CVCL_1292) or Cell Lines Service (T24; 300352/RRID: CVCL_0554, Eppelheim, Germany) or re-typed (253 J, RRID: CVCL_7935; RT4, RRID: CVCL_0036; RT112, RRID: CVCL_1670; 486 P, RRID: CVCL_7940; 5637, RRID: CVCL_0126; 639 V, RRID: CVCL_1048; EJ28, RRID: CVCL_5983; J82, RRID: CVCL_0359; KU1919, RRID: CVCL_1344; SW1710, RRID: CVCL_1721). All cell lines were grown in DMEM medium (Life Technologies 11320033, Carlsbad, USA) supplemented with fetal bovine serum 5% (v/v; Biowest S1810-500, Nuaillé, France), L-glutamine 2 mM (Merck 59202 C, Darmstadt, Germany) and penicillin/streptomycin 1% (v/v; Life technologies 15140122, Carlsbad, USA). Cell detachment was performed using Accutase (Thomas Geyer 8000921, Berlin, Germany).

## Proliferation assay

The MTT assay kit (ATCC 30-1010 K, Manassas, USA) was used with cells being seeded at 4,500 per well in 90 µL quintuplicates in separate 96-well plates for each time point. Cells were incubated for 24 h before the medium was replaced with fresh medium containing the inhibitor in DMSO at different concentrations specific to the expected $EC_{50}$ of the inhibitor (with the exception of cisplatin, well soluble in PBS). Controls were treated with DMSO with the same concentration (<0.1% v/v). Inhibitors were apicidin (Merck A8851, Darmstadt, Germany), PI-103 (Merck 528100), Flubendazole (Selleck Chemicals S1837, Planegg, Germany), SB225002 (Merck SML0716), KU 0063794 (Merck SML0382), calyculin A (Merck C5552, has to be stored at −20 °C), vincristine sulfate (Selleck Chemicals S1241), mitomycin C (Merck 10107409001) and cisplatin (Merck 232120). Depending on their intrinsic proliferation rates, cell lines were grown for 0/48/120/144 h (HT1376, RT4) or 0/24/48/72 h (all other cell lines).

## Basic statistical analyses and visualization

All statistical analyses and data visualizations were performed in Python (2.7.17 and 3.9.9) using the basic packages numpy 1.16.1, matplotlib 2.2.4, seaborn 0.9.1, and pandas 0.24.2. Principal component analysis was performed as implemented in scikit-learn version 0.20.4; statistical tests as implemented in scipy's (version 1.2.2) stats module. Sankey plots were drawn with a modified version of pySankey 0.0.1[106].

## LC-MS/MS data preprocessing and normalization

TMT set-specific data from the identification software was reevaluated on peptide level: Only unique peptide spectrum matches (PSM) were used. Each unique PSM was divided by the set standard intensity (always tag 134 N). Ratios of all unique PSMs per protein were then averaged. The sample values from all sets were then merged and the data normalized using Normics$_{median}$[107] with 100% sample coverage as input set (zero intensities accepted) and the 40% most invariant proteins used as normalization subset. One outlier sample with 8-fold lower (non-normalized) median intensity compared to all other samples was excluded.

## Sample-specific expression profiles

To fully exploit the value of our paired data, we normalized each tumor sample to its corresponding healthy mucosa (all data were $\log_2$-transformed). To assign significance, we then compared each protein $j$ of this relative sample $a_i$ to a pseudo-individual null distribution $H(i)$. This distribution was estimated by comparison of the respective healthy sample to all other healthy samples within the same TMT set. To account for the different variances of different PSM levels, we filtered $H(i)$ with a binning function $B(p, p_j)$, which included only the part of the null distribution that contained proteins with a similar number of PSMs:

$$B(p,p_j) = \begin{cases} 1, \text{ if } \frac{1}{2}p_j < p < 2p_j \\ 0, \text{ else} \end{cases} \quad (2)$$

with $p$ being the number of PSMs and $p_j$ the number of PSMs for protein $j$ (Fig. 4b). We computed the estimate of the standard deviation of the filtered null distribution and normalized $a_{i,j}$ with it. Using this normalized value $\tilde{a}_{i,j}$ as test statistic, we computed the p-value as $1 - \int_{-\infty}^{|\tilde{a}_{i,j}|} \mathcal{N}(0,1)$ (essentially the simple measure of how likely the observation of a value $\tilde{a}_{i,j}$ would be given the null distribution). Of note, p-values were not corrected for multiple testing, as individual independent tumor samples were tested against individual null distributions. Conclusions on cohort level were based on the frequency of an observation, providing quasi-repetition of the observation.

## Differential expression analysis

Differential expression between grouped protein intensities was performed using the Mann-Whitney U test and correcting the resulting p-values by the Benjamini-Hochberg method. The confidence interval of the difference in means was calculated using the tconfint_diff function of the statsmodels 0.9.0 package. All proteins with ≥20% non-missing values across the samples of the respective group. For the healthy-vs.-tumor comparisons out of 3079 differentially expressed proteins (B-H adjusted p < 0.05) 2997 could be mapped to proteins in the STRING database (v.12.0[108]. Then, functional enrichment analyses[109] of the gene ontology biological processes (v.42.442[110,111]) were performed.

## Unsupervised clustering

Non-negative matrix factorization was performed using the nimfa package (1.16.1) by Žitnik and Zupan[112]. The lowest 25% of the Normics ranking list computed as part of the normalization (so the most variant proteins) were used as input. Clustering was performed from k = 2 to k = 10, parameters were max_iter=k*100, update = 'divergence', objective = 'div', n_run=100, seed = 'random_vcol'. Dendrograms were calculated using scipy's cluster.hierarchy.linkage function with mode 'average'. Cluster relevance was computed as implemented in nimfa based on Kim and Park[113].

## Survival analysis and regression

Kaplan–Meier and Cox proportional-hazards regression models were fitted as implemented in the lifelines library (version 0.26.0)[114]. pTNM and PAULA clusters were treated as quasi-continuous ordinal variables ordered by their established prognostic hierarchy (pTNM) or survival stratification (PAULA).

## Pathway analysis

To examine activation of the main oncogenic pathways, the PROGENy tool was used[34] in its python implementation based on the modules decoupler 1.1.0[115] and omnipath 1.0.5[116]. Gene/protein set enrichment analysis was performed using GSEApy 0.10.8[117] with the 2018 gene ontology terms for biological processes.

## Feature selection

For all classifiers the filtered input training data was screened for correlated features and only a single (random) feature retained for a Spearman rank correlation coefficient ≥0.7. For IHC and cell line classifiers, support vector classification (SVC) was used as estimator and the training data resampled for 50 runs. During each of these runs 20% of the data were used for validation and recursive feature elimination with five-fold cross-validation as implemented in the sklearn library's[118] feature_selection.RFECV function. The occurrence of each feature in these resulting selections was counted and all features ordered by their frequency. For the cell line classifier with subsequent prediction of data with unknown truth, accuracy was estimated by resampling, training and testing within the data with known classes and the optimal number of features selected (75). For the prediction of mRNA subtypes only few proteins/transcripts remained after filtering (see below)−of these all 75 features were used that also occurred in the actual PAULA NMF cluster subset.

## mRNA classifier and proteogenomic contextualization

To predict the cluster classes of the external mRNA dataset (Robertson et al.[9] retrieved though the cBioPortal[119]), the intersection set of mRNAs and proteins were filtered by sufficient (significant) correlation with a Spearman rank correlation coefficient ≥0.75 in the comparative dataset by Wang et al.[35]. The data from both training (protein) and prediction (mRNA) data was then transformed by quartiles, with the 2nd and 3rd quartile being combined (restricting the data structure to low−medium−high due to the lack of information on the exact quantitative relationship between the two data types, which cannot be extrapolated universally from the comparative data). A gaussian kernel was used for SVC with the predict_proba method from the sklearn SVC class to estimate the prediction probability for the different cluster classes.

Mutations were filtered by non-synonymity (missense and nonsense) and filtered for a COSMIC variant count ≥5[120] using OpenCRAVAT 2.4.2[121]. Copy number variants (CNV) were from the GISTIC v.2 analysis by[9] and filtered by COSMIC gene occurrences ≥5 with chromosome positions from Ensembl.org (GRCh38.p14)[122].

## Cell line classifier

Cell line mRNA expression data was obtained from the Cancer Cell Line Encyclopedia[123] as FPKM, accessed via Expression Atlas (identifier E-MTAB-2770)[124]. Available proteomic data overlapped only partially with the cell lines investigated in vitro[125] and was used for correlation filtering (Spearman's correlation coefficient <0.75 excluded) of the more comprehensive transcriptomic coverage. For the transformation of the filtered transcriptomic data in analogy to the mRNA validation classifier, all available bladder and urothelial cancer cell lines except for squamous cell carcinoma were included (so more than the experimentally used cell lines) to maximize the coverage of the abundance ranges and to ensure similar relative distributions of the transcript quantiles compared to the proteomic data. A Gaussian kernel was used for SVC.

## IHC classifier

As feature selection was the sole objective of the classifier for proteins in common IHC use, a simpler approach using a linear SVC kernel was used. Resampling was performed for 100 runs. A three-tiered quartile transform of the LC-MS/MS protein data was used in analogy to the other classifiers to match the IHC levels.

## Drug repurposing

The connectivity map (CMap) platform was used[40] with correlation-filtered qualitative protein signatures as input (as CMap profiles are transcriptome-based; filtered in analogy to the classifiers). The latter were obtained from the sample-specific expression profiles. With CMap-available and proteomic data overlapping only partially,

n = 180 samples could be mapped. Connectivity scores ≤−90.0 were considered as relevant predictions.

## Cell line drug sensitivity

Cell line-specific MTT proliferation data was fitted by the Hill function using a non-linear least-squares estimator as implemented in the scipy.optimize function curve_fit. Additional published data was downloaded via DepMap[126] and PharmacoDB[127,128] from the datasets GDSC1 and 2[129], CTRPv2[130], PRISM[131] and gCSI[132].

## Reporting summary

Further information on research design is available in the Nature Portfolio Reporting Summary linked to this article.

## Data availability

All LC-MS/MS raw data has been deposited at the PRIDE repository[133], available under proteomeXchange.org with the identifier PXD041268 (cohort data), PXD041733 (method experiments) and PXD041873 (method experiments setup #1). Postprocessed data is available via an interactive web interface at www.cancerproteins.org. Source data are provided with this paper. Several publicly available datasets were used in this study: TCGA bladder cancer RNAseq, mutation and CNV data (www.cbioportal.org/study/summary?id=blca_tcga)[9]; mRNA/protein correlation data for TCGA reclassification (www.embopress.org/doi/full/10.15252/msb.20188503)[35]; cell line mRNA expression data from the Cancer Cell Line Encyclopedia (www.ebi.ac.uk/gxa/experiments/E-MTAB-2770/Results)[123]; corresponding protein abundances (www.cell.com/cms/10.1016/j.cell.2019.12.023/attachment/3709dedc-3a01-4e1d-ab4c-82597295c5d2/mmc2.xlsx)[125]; COSMIC data for mutation filtering (via www.opencravat.org)[120]; chromosome positions in GRCh38.p14 (www.ensmbl.org)[122]. Source data are provided with this paper.

## Code availability

The evaluation code is available online (https://github.com/ffdre/PAULA; https://doi.org/10.5281/zenodo.10937560)[134].

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

## Acknowledgements

F.F.D. thanks the Else Kröner-Fresenius-Stiftung for funding this work (2018_A84), Dr. Annika Fendler, Berlin, and Professor Christine Sers,

Berlin, for their valuable comments and advice, and Dr. Miriam Dreßler, Berlin, for her invaluable advice and support. F.F.D. is a participant in the BIH Charité Clinician Scientist Program funded by the Charité—Universitätsmedizin Berlin and the Berlin Institute of Health at Charité (BIH). H.S. is supported by the Deutsche Forschungsgemeinschaft (INST 337/15-1, INST 337/16-1, INST 152/837-1, INST 152/947-1 FUGG and SCHL 406/21-1).

## Author contributions

Conceptualization, funding acquisition, formal analysis, software, visualization, supervision, project administration, writing—original draft: F.F.D.; methodology: F.F.D., A.V.; investigation: F.F.D., A.V., S.H., P.M., M.S., C.K., T.G., M.H., F.D., D.S.; data curation: F.F.D., A.V., T.G., C.K., H.V.; validation: F.F.D., S.H.; resources: F.F.D., U.W., A.M., H.S., R.Z., S.P., A.S., T.G., P.W., M.K.; writing—review & editing: F.F.D., A.V., R.Z., U.W., A.M., S.P., H.S., A.S., T.G., C.K., A.O., D.S.S., C.G., J.S., M.C.R, P.W., M.K., J.K.

## Funding

## Competing interests

The authors declare no competing interests.
