## [Peer Review File · Nature Communications]

Reviewers' Comments:

Reviewer #1:

Remarks to the Author:

In the current study "Proteomic Analysis of the Urothelial Cancer Landscape" the authors (who remained anonymous) performed proteomics analysis of a heterogenous cohort of archival bladder cancer specimens (FFPE sections with microdissections for enrichment). The authors identify five proteomics molecular subtypes (they call the PAULA clusters) with prognostic value. While proteomic subtypes were related to histopathologic stages they did provide some additional value. How this would be used in the clinic and how it would validate in independent samples at the proteomics level is unclear. Using a connectivity map approach the authors also suggest some PAULA specific vulnerabilities, which they attempt to validate using UC cell line models.

The patient cohort used in this study is solid and the proteomics approach which the authors called Surface Maximization and Reduction of Dimensionality (SMRD) seems of good quality. Where the paper falls short is on utilization of these data. The manuscript often felt convoluted and was very hard to follow (as a reviewer that uses clinical proteomics for over 20 years I often was unclear what the authors actually did). At this stage the paper is not suitable for Nature Communications (with some major refocus and rewriting it could be). There are many comments I could leave, but I will focus on some major suggestions that might help the authors to modify the manuscript.

Comments:

- 1) I felt the proteomics method development in the beginning of the paper very distracting. The authors effectively modified a FFPE tissue proteomics protocol that might give slightly better coverage (frankly speaking I am not even convinced this would be better than a SP3 or TFE-based protocol).
 - a. All these different methods that are supposedly compared at very convoluted, not even mention they are all using slightly different mass spec set-ups. I fail to see the relevance. This section would be much better served in an independent, small methods paper in a proteomics journal.
 - b. The authors also make some strange comments that archival FFPE tissues have been rarely used in clinical proteomics (I can think of many studies, some from our group and many others). Yes CPTAC tissues are specifically collected for this consortium and cohort quality and annotation is often debatable.
 - c. The normalization method, that was actually hard to follow, could also be in a small methods paper. It is distracting from a resource/clinical proteomics paper.
- 2) Along these lines and related to the method. The authors multiple times talk about detection of surface antigens for ADC (or immunotherapy development). I found this also extremely distraction since the entire mining strategy is not even focused on surface antigens (it would require a much more focused approach) and at the end of the day nothing is done with immunotherapy in the paper. This is oversold and not relevant in the current manuscript (not even mentioning the many caveats I could come up with). There are also much better methods to detect cell surface proteins (glycol-capture for example).
- 3) Totally unclear why integrin beta 1 was chosen for validation. This is likely one of the most abundant and commonly expressed cell surface adhesion molecules. If the authors want to sell this approach as a pipeline for discovery of novel immunotherapy targets then more relevant targets should be selected.
 - a. Why would cofilin-1 be a good loading control? Even the immunoblot signals in Fig 5B do not look very convincing. Likely a total Coomassie blue stain of the membrane would be a better loading control.
- 4) I found the validation of the PAULA groups by the authors TCGA RNA-seq analysis and the IHC very weak.
 - a. The authors are correct that proteomics and RNA are often only poorly correlated. Which makes it often challenging to use RNA for independent validation of proteomics data. Here the authors mined down TCGA data to select a super small subset of transcripts (75 from what I see in 5D). It's almost a circular argument and the provided validation in 5E seems less than convincing. Personally, I feel that this validation by RNA is not required here and the authors lost an opportunity to utilize TCGA for additional insights. It would have been much more interesting to see if PAULA clusters are associated with some genomics, epigenomics and methylation features of these tumors that were sequenced by TCGA.

- b. I found the poor validation with IHC somewhat more concerning.
 - i. Unclear how the 9 markers in 5F were actually selected
 - ii. How do we even know the antibodies are good (the most challenging feature of validating proteomics is poor quality of antibodies).
 - iii. The two methods nevertheless provide complementary data. Proteomics provides a average intensity of a proteins from a lysate of dissected tissue while ICH provides spatial or even cellular/sub-cellular data. These are not always easily mergeable.
 - iv. While I am a big supporter of proteomics I found it a bit confusing that the authors considered their proteomics data as the ground truth for these comparisons. How do we know that is true. More importantly, if neither RNA nor IHC could be used to validate these patient subgroups, how would this ever be used in the clinic? At this point one could almost ask the authors to validate by proteomics or targeted proteomics in additional samples. It becomes complex!
- 5) How were the UC cell lines selected for the validation of drug sensitivities.
- a. Why can the cell lines be classified into PAULA subgroups by transcriptional profiles if this only works weakly for tissues?
 - b. 4 of 7 predictions could be confirmed? This is almost random chance.
 - c. Fig 6C hard to interpret (not very convincing).
- 6) I was not able to find the datasets in Pride using the PXD accession provided?
- 7) The link to the cancerproteins.org web-interface is not very useful in its current form. It seems that only single proteins can be searched. Better functionality would be required.
- 8) The supplemental tables lack description and are formatted in a way that it is unclear what we are looking at. It's not hard to format and describe a proteomics supplemental table!

Overall, a potentially interesting paper that could benefit from better focussing. Some of the validations (RNA, IHC were not very convincing in current form). The same was felt for the drug sensitivities. The section on ADCs is totally oversold and at the moment not relevant

Reviewer #2:

Remarks to the Author:

In this article, the authors conducted a proteomic analysis of a urothelial cancer cohort consisting of 196 patients with a broad spectrum of different UC types. This study provided valuable resources for urothelial cancer research, enabling both biomarker and novel therapeutic target discovery, as well as mechanism studies. The authors developed a protocol optimized for FFPE samples. While additional benchmark analyses can be performed, the successful sample preparation protocol for proteomic analysis of FFPE samples can potentially reduce study costs and enable retrospective studies of routine diagnostic samples.

1. TMT Normalization: The authors employed a custom algorithm for protein normalization and quantitation. However, the normalization algorithm neither normalizes for the total amount of material in a channel nor across TMT sets. The authors reported a predominant downregulation of proteins in tumor vs. normal samples, which could be biased by the lower total protein content in tumor vs normal samples. Failing to perform cross-plex normalization makes feature quantities from different TMT sets incomparable to each other and could affect downstream analyses, including survival and association analyses, and potentially unsupervised clustering if proper normalization wasn't applied. The authors may consider exploring TMT-integrator and PAW proteomics pipeline (https://github.com/pwilmart/PAW_pipeline). The authors could also demonstrate that their normalized protein quantitation is similar to well-recognized TMT normalization/quantitation algorithms, of which a cross-plex normalization is employed, and that biases were not introduced to their analysis.

2. TMT labeling: The article mentioned that either 5 or 10 µg samples were used depending on the total amount of peptides. This should be clarified, and it should be demonstrated that the total protein/peptide amount does not introduce bias. It was mentioned in the same paragraph that samples were labeled in random order. Are samples from the same patients always placed in the same TMT set? Cross-plex comparisons are less reliable, even with proper cross-plex TMT normalization, and it needs to be demonstrated whether downstream analysis is biased by this randomization.

3. SMRD: The authors developed an protocol, SMRD, for sample preparation of FFPE samples for proteomics analysis and demonstrated its superiority over conventional sample preparation processes. It would be interesting to also compare SMRD on FFPE samples versus a conventional sample preparation method on fresh-frozen (FF) samples. The SMRD protocol involves a thermal lysis step with incubation at 90 °C for 60 min and 99 °C for 5 min. It remains unclear whether the thermal lysis introduces any artificial modifications, and it might be worthwhile to conduct an open search using MSFragger.

4. Statistical analysis: Differential expression analysis was conducted using the non-parametric Mann-Whitney U test. However, for paired data like this, a paired t-test or a more sophisticated mixed linear model would provide greater statistical power, even after excluding patients without the normal sample. The sample-specific expression profile analysis lacks clarity, and more details are needed on how p-values are computed. The equation requires further explanation, especially regarding the meaning of $B(p)$, and the first sentence below the equation does not make sense to me. Additionally, Figure 4C requires a more detailed explanation.

5. Figure clearance: The figures suffer from significant clarity issues. Please revise both the figures and their legends to ensure that all elements are clearly explained. It appears that '[1]' is used to indicate untransformed scales, but this notation seems redundant and causes confusion.

Minor issues:

Page 8 paragraph 3: precursor and the fragment ion mass tolerance to 10 ppm and 0.02 Assuming the unit for fragment ion mass tolerance is Da.
In the same paragraph, was contaminant proteins added to the database? How was decoy-database created?

Page 14 paragraph 2
What's the criteria of sufficient peptide intensities for quantitation?

Page 16 paragraph 3
I assume that the pTNM groups were converted to a dummy variable or factor variable to compute an overall hazard ratio. How was that done for PAULA clusters? How was the order of clusters decided?

Page 18 paragraph 2, "75 proteins with the highest overexpression rates across all samples"
Define "overexpression rates". I assume this refers to proteins with both elevated and lowered protein abundance, and maybe a more precise term can be used. And consider using the term "protein abundance" instead of "protein expression".

Page 20 paragraph 1 "Normalized ITGB1 intensities showed strong positive correlation with the mass spectrometric abundances"
P-value and correlation method is missing.

Figure-Specific Issues:

Figure 1C: Clarify why different boxes appear to have different alpha values.

Figure 1E: Explain the meaning of " $n=x/x/x/x/x$ " in the second line of each box.

Figure 1H: Clarify the meaning of "share."

Figure 4C: Provide further clarification for this panel.

Figure 5E: Ensure that the p-value is visible and not obscured by the legends.

Reviewer #3:

Remarks to the Author:

The authors profiled the proteomes of 434 samples from 196 patients with urothelial bladder cancer. They proposed a robust proteomic classification with additional prognostic information independent from histopathological groups and showed that proteomic subtypes were predictive

for these drugs by using a comprehensive cell line panel.

Having said that there are significant problems with the current manuscript:

Major

Q1. My major concern is with the sampling procedures. Starting from 196 patients, the authors analyzed 242 tumor samples and 192 paired healthy tissues, obtained by manually dissecting using the H&E template to yield a total sample area of ca. 1 cm². Does healthy tissue include only normal urothelium? Due to the tissues of normal urothelium being very tiny, how do the authors ensure that only normal urothelial tissues are scraped and not mixed with tumor tissue? I would like to receive some more details on this procedure and, most important, evidence should be provided that the scraped tissues correspond to the tissue annotated by the pathologists.

Q2. How about the tumor purity of these tumor samples? Is there any difference in tumor purity among these proteomic subtypes which may influence the molecular features?

Q3. The authors used the SMRD protocol to process samples. Is this SMRD protocol being used for the first time? Has it been reported in other published literature?

Q4. The author obtained 242 tumor samples from 196 patients, which means that multiple tumor samples were taken from one patient. The author conducted protein clustering on the 242 tumor samples. Does this also mean that a patient can be classified into different protein clusters? For example, a patient can be classified in the PAULA I group with a good prognosis and also be classified in the PAULA III group with a poor prognosis. In this case, are these patients considered to have a good or poor prognosis? This is confusing and needs clarification.

Q5. What are the criteria for the definition of overexpressed proteins? i.e. "Thymosin beta 10 (TMSB10; Fig. 4G) – the only protein also found amongst the five upregulated proteins from the standard differential expression analysis (Fig. 4A) – was overexpressed in 52 % of the >pT2 samples, and in 15 – 30 % of PAULA I, IIa, IIb, and III samples." What is the cutoff for the definition of TMSB10 overexpressed in 52 % of the >pT2 samples? "The luminal markers SRC, RAB25, E-cadherin, and ERBB2 demonstrated only mild overexpression in PAULA I samples (up to 13 %), while the basal-associated EGFR was overexpressed in 22 % of PAULA III samples (2 % in PAULA I)." What about the mild overexpression?

Q6. Surprisingly, the number of upregulated proteins in tumors (5 proteins) was far less than the downregulated proteins (177 proteins) in Fig. 4A. However, tumor cells should express more proteins than normal urothelial cells to perform more functions. Should be clarified and discussed.

Minor

Q7. Page 7, paragraph 3, and paragraph 5. "The samples covered all stages of UCC...The five clusters mirrored the tumor-biological spectrum of UCC". What is the UCC? Moreover, the authors only collected patients with urothelial bladder cancer and it is recommended to use more accurate terminology in the title of "Proteomic Analysis of the Urothelial Cancer Landscape", as urothelial carcinomas encompass both urothelial bladder cancer and upper tract urothelial carcinoma.

Q8. Page 11, paragraph 3. "Of the seven predictions, four could be confirmed (Fig. 6E)." Where is Fig. 6E?

Q9. There are a few claims that are overstated. For example, the authors say "Normalized ITGB1 intensities showed a strong positive correlation with the mass spectrometric abundances (Fig. 5A).", but the correlation coefficient of 0.5 would be considered only a moderate value.

Q10. The 95% confidence intervals should be presented in Fig 3A for principal component analysis.

Q11. The method for survival analysis needs to be supplemented. For instance, what is the cutoff point for the selected sample in Figure S7?

Point-by-Point Response for Submission NCOMMS-23-41307-T

Proteomic Analysis of the Urothelial Cancer Landscape

Reviewer #1

In the current study “Proteomic Analysis of the Urothelial Cancer Landscape” the authors (who remained anonymous) performed proteomics analysis of a heterogenous cohort of archival bladder cancer specimens (FFPE sections with microdissections for enrichment). The authors identify five proteomics molecular subtypes (they call the PAULA clusters) with prognostic value. While proteomic subtypes were related to histopathologic stages they did provide some additional value. How this would be used in the clinic and how it would validate in independent samples at the proteomics level is unclear. Using a connectivity map approach the authors also suggest some PAULA specific vulnerabilities, which they attempt to validate using UC cell line models.

The patient cohort used in this study is solid and the proteomics approach which the authors called Surface Maximization and Reduction of Dimensionality (SMRD) seems of good quality. Where the paper falls short is on utilization of these data. The manuscript often felt convoluted and was very hard to follow (as a reviewer that uses clinical proteomics for over 20 years I often was unclear what the authors actually did). At this stage the paper is not suitable for Nature Communications (with some major refocus and rewriting it could be). There are many comments I could leave, but I will focus on some major suggestions that might help the authors to modify the manuscript.

We thank the reviewer for their appreciation of our methodological work and the many detailed corrections and suggestions on how to improve our manuscript. We tried to include all aspects mentioned and rewrote and restructured the manuscript accordingly – as far as this could be done in accordance with the other reviewers’ comments.

Major Comments

- 1. I felt the proteomics method development in the beginning of the paper very distracting. The authors effectively modified a FFPE tissue proteomics protocol that might give slightly better coverage (frankly speaking I am not even convinced this would be better than a SP3 or TFE-based protocol).*

We agree that the method optimization in the beginning is a bit separate from the rest of the manuscript but hope that we can explain our reasoning and adjust the manuscript with the following aspects (please also see below). As a matter of fact, our protocol already includes an SP3 part to clean up the peptides in setup #2 and the final cohort. We clarified the respective method statements:

“In analogy to the SP3-like clean-up in setup #2 the peptide samples were cleaned on a C18 Hyper-sep plate with 40 µL bed volume”

Within this modification of the SP3 protocol we compared our optimized method to an approach without our optimization steps (still including the SP3 clean-up, of course). For this strictly controlled setup we observed relevantly increased identification rates of cytosolic and membrane proteins. While we agree that this can probably further be optimized and that it might be beneficial to include carboxylate beads already at the time of the acetone precipitation (which we are currently working on), we found it difficult to combine it with direct trypsinization (modified in our protocol), which we deem essential to extract all proteins efficiently from FFPE tissue. In essence, we can show in direct comparison that our approach (including an SP3-based clean-up) is superior to regular direct trypsinization, and we have thus reason to expect it to be so in different, even more advanced setups. To underline the potential for further improvement we included a statement in the limitations paragraph of the discussion:

“Concerning an ideal proteomic sample preparation strategy, we observed that balancing deep sample extraction and sample loss is pivotal. Our proposed SMRD protocol could potentially further be refined with regard to the latter, e.g. by optimizing binding of the cell fragments to carboxylate paramagnetic beads upfront.”

- a. *All these different methods that are supposedly compared at very convoluted, not even mention they are all using slightly different mass spec set-ups. I fail to see the relevance. This section would be much better served in an independent, small methods paper in a proteomics journal.*

We thank the reviewer for their suggestion. Indeed, we had contemplated publishing the protocol in a separate methods paper. However, we are trying to make a point for the value and application of clinical proteomics, which needs to deal with suboptimal and small samples. This is of special relevance in UC, where samples are often degraded (due to inflammation) or of limited material (for instance small areas of focal invasion). When we planned the study, we found the usual analysis depth of LC-MS/MS insufficient for the reliable identification and quantification of the relevant oncogenic driver proteins (many of which are membrane proteins). Back then, we decided to optimize sample processing first, especially so to enable the routine-ready measurement of real-world FFPE tissue. We stress this “real-world” aspect here and in the manuscript, as – in our experience – there are considerable protein quality differences between fresh-frozen tissue, FFPE tissue with standardized tissue size and fixation (as in a prospective study) and the actual diagnostic material, for which tissue size and fixation are neither standardized nor known.

Since another reviewer has commented positively on the SMRD protocol, we would like to keep it in the manuscript. However, we clearly need to better outline the interplay between this methodological part and the rest of the manuscript, which is why we revised the respective paragraphs.

For instance, in the introduction:

“Especially routine diagnostic formalin-fixed paraffin-embedded (FFPE) tissue adds further complexity and reduces the analytical depth due to heterogeneous processing and fixation artifacts. [...]”

We further advance pre-analytical sample processing to increase the analysis depth of clinical FFPE specimens and to improve robustness across setups and the heterogeneous samples of the UC landscape.”

In the beginning of the results section:

“To analyze a comprehensive all-stage UC cohort from routine FFPE tissue and to measure target specificity quantitatively, we developed an optimized protocol to increase analysis depth for standard clinical samples with heterogeneous tissue types, fixation, and processing.”

In the SMRD subsection:

“To cover the clinical UC spectrum, sufficient performance even with small amounts of input material needs to be ensured, which is typically obtained after minute dissection, e.g., for focally invasive areas. Also, analysis depth is particularly relevant as key oncogenic proteins are of comparably low abundance and often are transmembrane proteins. Liquid chromatography-coupled tandem mass spectrometry (LC-MS/MS) of complex biological samples is negatively affected by quantitative imbalances within the sample. These are caused by highly abundant proteins, such as smooth muscle or extracellular matrix (ECM) proteins. Membrane proteins in turn, key players in oncogenic signal transduction and metastatic processes, are even less accessible due to their reduced solubility, co-localization with structural and ECM proteins, and geometric abundance imbalance [...]”

We sincerely hope that these corrections find the reviewer’s approval and allow us to keep this method improvement in the paper.

Concerning the different setups: We compared the same protocol and controls with several setups, again all to show the methodological robustness of our approach across different labs and setups. So the “slightly different mass spec set-ups” were in fact on purpose to show this. We included a clarifying statement in the results section:

“We evaluated SMRD performance against direct trypsinization in a controlled setting and used different LC-MS/MS setups and quantitation methods to ensure maximal robustness.”

- b. *The authors also make some strange comments that archival FFPE tissues have been rarely used in clinical proteomics (I can think of many studies, some from our group and many others). Yes CPTAC tissues are specifically collected for this consortium and cohort quality and annotation is often debatable.*

We apologize for the wording of our statement (we meant only the urothelial cancer studies). We removed this paragraph, as it is not necessary and potentially misleading.

c. The normalization method, that was actually hard to follow, could also be in a small methods paper. It is distracting from a resource/clinical proteomics paper.

We thank the reviewer for this point and agree that this aspect is better covered in an independent small technical brief. We have therefore only included the description of the method in the methods section but removed the respective subsection in the results. We also adjusted the discussion accordingly.

2. Along these lines and related to the method. The authors multiple times talk about detection of surface antigens for ADC (or immunotherapy development). I found this also extremely distraction since the entire mining strategy is not even focused on surface antigens (it would require a much more focused approach) and at the end of the day nothing is done with immunotherapy in the paper. This is oversold and not relevant in the current manuscript (not even mentioning the many caveats I could come up with).

We thank the reviewer for this critical point. We agree that the specific mention of ADCs is misleading as we did neither develop a new ADC nor are the biomarkers we describe exclusively targetable with ADCs (other mechanisms could also be used). The implicit reason was that ADCs are the most common class of targeted therapies that a) depend on quantitative tumor specificity (in contrast, e.g., to qualitative mutations) and b) that have already demonstrated clinical value in UC (Enfortumab Vedotin very recently even in an all-comer first-line setting for advanced and metastatic disease). Nonetheless, one rationale of our study is closely linked to this therapeutic principle of targeting a descriptive protein biomarker (i.e. not a functional inhibition, so in contrast to, e.g., the *KRAS* p.G12C mutation-targeted inhibitor Sotorasib). This approach is not only effective in UC but also in several other entities such as breast cancer (ERBB2, Trastuzumab Deruxtecan), gastric cancer (CLDN18.2) or uveal melanoma (PMEL, Tebentafusp). Still, a systematic omics scale analysis of the most suitable target candidates with the highest tumor specificity and overexpression rates is currently lacking for UC (and most likely for other entities, too), instead targets are chosen based on (rare) recurrent genomic alterations (such as Enfortumab Vedotin against NECTIN4) and/or based on very limited immunohistochemical data (Enfortumab Vedotin and Opportuzumab Monatox against EpCAM). This quantitative foundation needs to be proteomic, and it needs to sufficiently cover membrane proteins. We set out to fill this gap and to provide quantitative proteomic data across the diverse UC landscape. We extensively revised the manuscript to better explain this rationale and to tone down potentially misleading statements. We thank the reviewer again for highlighting this important translational aspect that surely needed better communication and wording.

Amongst other changes, the abstract now reads:

“Urothelial bladder cancer (UC) has a wide tumor-biological spectrum with challenging prognostic stratification and relevant therapy-associated morbidity. Most molecular classifications relate only indirectly to the therapeutically relevant protein level. We improve preanalytical preparation of clinical samples for proteome analyses and characterize a cohort of 434 samples with 242 tumors and 192 paired normal mucosae covering the full range of UC. We evaluate sample-wise tumor specificity and rank biomarkers by target relevance. Antibody drug conjugates (ADC) targeting cell surface proteins promise better disease control and have recently advanced to the clinic. However, systematic all-stage approaches to identify and quantify potential targets are lacking. Here, we propose SMRD, a new protocol for deep proteomic profiling of routine clinical samples and characterize a comprehensive cohort of 434 samples, covering the full range of UC. We identify robust proteomic subtypes with prognostic information independent from histopathological groups. *In silico* drug prediction suggests efficacy of several compounds hitherto not in clinical use. Both, *in silico* as well as *in vitro* data, indicate predictive value of the proteomic clusters for these drugs. These subtypes unite and expand transcriptional groups and are predictive of several new therapeutics, which we identified *in silico* and validated *in vitro*. We show that proteomics is necessary for personalized oncology and cannot be substituted by immunohistochemistry provide abundance and tumor specificity data for a large part of the UC proteome. Covering a large part of the proteome, we provide quantitative data on protein target specificity for the development of further ADC therapies (www.cancerproteins.org).”

In the results section:

“As one of the main objectives of our study, we set out to meet the need for robust quantitative data on the specificity of potential descriptive protein targets for ADCs and other targeting strategies. The dataset can be accessed directly at www.cancerproteins.org.”

In the beginning of the discussion:

“Prognostic and predictive stratification of UC patients is clinically challenging yet highly relevant [8]. Proteomics analyzes the cellular effectors and offers direct quantitation of ~~druggable~~ targets [74], which is vital information for tumor-specific targeted ADC therapies such as ADCs. We advance proteomic analyses towards clinical LC-MS/MS applications and report the prognostic and predictive PAULA subtypes. We provide comprehensive information about the tumor specificity of target expression and specificity for ADC and other targeted therapies, covering a large part of the proteome and all stages of UC.”

a. *There are also much better methods to detect cell surface proteins (glycol-capture for example).*

We thank the reviewer for addressing this important aspect and for bringing the glycol-capture protocol to our attention. In fact, we initially investigated several protocols for a better coverage of membrane proteins in FFPE diagnostic tissue, and we tried to cover all relevant physicochemical mechanisms. We assume that the glycol-capture protocol refers to Lee and Sun (2014). While we did not try the exact protocol, our comprehensive, hierarchical comparison of different existing methods as well as the development of new optimized protocols included a similar approach. Our findings with this and other techniques have been summarized as supplementary Note 1.

Revision Figure 1: Standard lipid phase extraction with fresh cells and diagnostic-grade formalin-fixed and paraffin-embedded (FFPE) tissue. Reproduction of Fig. 1A from the supplementary Note 1.

The glycol-capture protocol essentially combines two enrichment steps: First, a membrane “harvest” or enrichment by differential centrifugation is performed, followed by oxidization of diol groups (after protein digestion), which enables binding to a resin matrix with subsequent clean-up and enzymatic freeing of the N-bound glycopeptides by PNGase F. Apart from the apparent limitations that only glycosylated peptides are enriched (which does not comprise all membrane proteins nor (unique) peptides thereof) and that within this subset only N-linked, not O-linked glycopeptides are analyzed, there are further problematic aspects when applying this protocol to both clinical and FFPE tissue. Before addressing those related to the different properties of FFPE tissue as opposed to the use of fresh cells, a very basic yet relevant hurdle is already the amount of input tissue. The glycol-capture protocol requires 10^8 cells, which translates (under the assumptions of a pure tumor cell block with minimal connective tissue, a cell size of 25 μm , spherical volume and perfect packing with the theoretically optimal density of 0.74) to about 1 cubic centimeter (mL) of tumor tissue. Given the necessity to dissect only areas with sufficiently high tumor purity and the relevant tumor biology (e.g., (micro-)invasion) we already processed a total of four thousand sections for our cohort to obtain around 1 cm^2 with a thickness of 20 μm per sample. We would have had to scale up these efforts by a factor of 500 to 1000 to acquire sufficient input material, which is unfeasible. We demonstrated this aspect for another enrichment protocol based on lipid phase extraction. We reproduce Fig. 1A of our

supplementary Note 1 here for the reviewer's convenience, where we compared a fresh cell-based protocol with only 10^6 cells as input material to a realistic amount of diagnostic FFPE tissue. While the protocol worked for the fresh cells, sample loss was too high for our actual tissue (Fig. R1). Next, we also tried a differential centrifugation protocol, equivalent to the respective part of the glycol-capture protocol. We have created a new panel to the figure in the our supplementary Note 1 (Fig. R2) which shows our results (immunoblot for four plasma membrane and cytosolic/mitochondrial markers): Apart from (again) relevant sample loss, there is no sufficient separation of the different fractions. We went even further and tried density gradient centrifugation with similar, insufficient results. From our experience with FFPE tissue in general and our second, unsuccessful protocol development (please also see our supplementary Note 1), during which we tried to reconstitute the membrane proteins/parts into artificial liposomes for enrichment, we must report that FFPE tissue remains fundamentally different from fresh (frozen) tissue, even after crosslinking reversal (which we have previously investigated and optimized separately). This is mirrored by the harsh conditions for crosslinking reversal and leads to the last limitation of the glycol-capture protocol. The multitude of formalin-induced modifications (e.g., Kamps, Hopkinson et al. (2019)), which will neither be completely reversed nor brought back to their exact initial molecular configuration, would also affect the glycans' diol groups, rendering sufficient oxidization and linking to the resin unlikely.

Revision Figure 2: Differential centrifugation of FFPE extracts and immunoblot quantification of the relative abundances of compartmental markers. The protocol was performed in duplicate replicates. Our apologies for the tear in the first lane, which, however, does not affect the basic conclusions from this experiment: There is no relevant enrichment, whereas there is relevant sample loss.

We have tried to highlight the relevant differences between FFPE and fresh (frozen) tissue and the difficulties that arise in the application and development of any new protocol. Nonetheless, we are convinced that efforts need to be made to tailor proteomics to this scientifically valuable material. While we agree that our protocol can potentially be improved further (please also see our response to your first point and our added statement in the limitations), we made such an effort and spent considerable time and resources on a systematic exploration of possible preanalytical improvements of FFPE tissue proteomics with a special emphasis on membrane proteins (overall, we ran over one hundred methodological experiments). Within this challenging context and the complexity as well as heterogeneity of diagnostic FFPE samples, we feel that our results are solid and of relevance.

3. *Totally unclear why integrin beta 1 was chosen for validation. This is likely one of the most abundant and commonly expressed cell surface adhesion molecules. If the authors want to sell this approach as a pipeline for discovery of novel immunotherapy targets then more relevant targets should be selected.*

We thank the reviewer for addressing this point and apologize for the perhaps misleading explanation in the manuscript. Immunoblotting was performed to validate the proteomic measurements technically, not to validate a new target. We chose integrin beta 1 for the very reasons mentioned, it is commonly expressed and a large transmembrane cell surface protein. Also, we and others (Adua, Arnal-Estape et al. 2022) have

previously used integrin beta 1 and the antibody clone 9699 and we knew that it could reliably be quantified by immunoblotting. We adjusted the respective sentence in the results section:

“To validate both SMRD sample processing and the TMT-based quantification, we chose the large membrane protein integrin beta 1 (ITGB1) as an exemplary benchmark protein, ~~which is also involved in cancer metastasis [62].~~”

- a. *Why would cofilin-1 be a good loading control? Even the immunoblot signals in Fig 5B do not look very convincing. Likely a total Coomassie blue stain of the membrane would be a better loading control.*

We have used cofilin-1 (CFL1) here and before as (generally accepted) loading control (e.g. www.bio-rad-antibodies.com/western-blot-loading-controls-antibodies). Total protein normalization implicitly assumes that only few proteins are differentially expressed between samples. For healthy-vs.-tumor comparisons we were not sure whether this assumption would hold, which is why we chose a more conservative approach with a housekeeping protein instead. Our LC-MS/MS data also shows constant abundance of CFL1:

Revision Figure 3: Abundance of CFL1.

Also, it goes without saying that we ensured a sufficiently linear dynamic range for both antibodies:

Revision Figure 4: Linear dynamic range of the immunoblot antibodies against integrin beta and cofilin-1.

As we already observed a positive correlation between immunoblot and LC-MS/MS data with our simple housekeeping approach, we think that more elaborate normalization strategies would not alter this result qualitatively (as one would need to assume some inverse correlation to get a false positive result).

4. *I found the validation of the PAULA groups by the authors TCGA RNA-seq analysis and the IHC very weak.*
- a. *The authors are correct that proteomics and RNA are often only poorly correlated. Which makes it often challenging to use RNA for independent validation of proteomics data. Here the authors mined down TCGA data to select a super small subset of transcripts (75 from what I see in 5D). It's almost a circular argument and the provided validation in 5E seems less than convincing. Personally, I feel that this validation by RNA is not required here and the authors lost an opportunity to utilize TCGA for additional insights. It would have been much more interesting to see if PAULA clusters are associated with some genomics, epigenomics and methylation features of these tumors that were sequenced by TCGA.*

We thank the reviewer for this very valuable suggestion, which helped us to better characterize our findings, leading to several new and relevant aspects. We expanded our analysis of the reclassified TCGA data as appended to the methods section:

“Mutations were filtered by non-synonymity (missense and nonsense) and filtered for a COSMIC variant count ≥ 5 [47] using OpenCRAVAT 2.4.2 [48]. Copy number variants (CNV) were from the GISTIC v.2 analysis by [9] and filtered by COSMIC gene occurrences ≥ 5 with chromosome positions from Ensembl.org (GRCh38.p14) [49]. Further CNV data including ABSOLUTE estimates were retrieved from gdc.cancer.gov [50].”

We created a new subplot as Fig. 5D and rearranged the figure to include this valuable data on associated mutations, copy number variants and methylation patterns as well as diagnostic parameters (Fig. R5).

Revision Figure 5: New subplot (Fig. 5D in the manuscript) with mutational, structural variant, and epigenetic information from the reclassified TCGA dataset. The 20 most frequent mutations as well as relevant copy number variations are shown; Small insets on the right visualize (per gene) the absolute frequency of mutations/amplifications/deep deletions (upper bar) and the relative frequencies per PAULA cluster (middle bar) or TCGA subtype (lower bar).

These analyses offered a more comprehensive comparison between our proteomic clusters and the transcriptomic TCGA subtypes. For instance, luminal tumors have previously been associated with *FGFR3* alterations. PAULA I samples contained almost all *FGFR3* mutations, while these were distributed across several TCGA subtypes.

We renamed and reorganized the respective paragraph in the results section and added our findings:

“Proteogenomic context with the TCGA dataset ~~External validation demonstrates prognostic stratification of MIBC samples~~

[...] We observed a general correlation of luminal (papillary) and basal (squamous) subtypes with PAULA I and III samples, respectively, while PAULA II subtypes were more heterogeneously associated (Fig. 5D, Suppl. Fig. S9). Interestingly almost all *FGFR3* mutations (which are a hallmark of luminal tumors) were contained in the PAULA I samples, whereas they were found more evenly distributed across luminal papillary, luminal, and neuronal subtypes in the TCGA data (similar frequencies were found in basal squamous and PAULA III samples). *TP53* mutations in turn were enriched in both PAULA III and PAULA IIa samples, two clusters that also showed subtle enrichment of hypomethylation clusters 4 and 1, respectively. Copy number variations such as amplifications in *PPARG*, *FGFR1*, and *NECTIN4* were more frequently observed in PAULA IIa samples, while neither *CDKN2A* (p16; as proposed marker for genomically unstable subtypes) gene alterations nor protein abundance

(Suppl. Fig. S8B) showed an association. Interestingly, the higher *NECTIN4* amplification rates in PAULA IIa samples were mirrored by higher protein abundances in PAULA IIa samples in our cohort (Suppl. Fig. S11). In our progression-free survival data (Fig. 3G), this molecularly distinct position of PAULA IIa was mirrored by diverging, reduced survival similar to PAULA III samples. In general, the re-classified TCGA samples showed survival stratification similar to our proteomic cohort (Fig. 5D-E).

Concerning further protein markers for the molecular (TCGA) subtypes, the luminal markers SRC, RAB25 [...]”

b. *I found the poor validation with IHC somewhat more concerning.*

i. *Unclear how the 9 markers in 5F were actually selected*

We followed a strictly data-driven approach to evaluate whether a panel of IHC antibodies could be used to subtype samples. We used a library of >200 antibodies for which diagnostic grade and validated assays were available (please see also our response to the next point). For these, we evaluated possible combinations *in silico* with our data. We then evaluated the top proteins by IHC (i.e. the proteins that showed the best classification performance in our LC-MS/MS data). We clarified the respective sentence in the results section:

“We then evaluated the top seven proteins by IHC on a random subset of n=54 samples.”

ii. *How do we even know the antibodies are good (the most challenging feature of validating proteomics is poor quality of antibodies).*

As a pathology-led study, we exclusively used diagnostic grade monoclonal antibodies for IHC, which were used in tissue-validated and diagnostically accredited assays (German DAkkS). We included a respective statement in the methods section:

“All antibodies were of validated diagnostic grade and all assays run and evaluated under diagnostic conditions.”

iii. *The two methods nevertheless provide complementary data. Proteomics provides a average intensity of a proteins from a lysate of dissected tissue while ICH provides spatial or even cellular/sub-cellular data. These are not always easily mergeable.*

We agree. At the same time, IHC should reflect the bulk quantities to some extent. We refer to our response to the next point.

iv. *While I am a big supporter of proteomics I found it a bit confusing that the authors considered their proteomics data as the ground truth for these comparisons. How do we know that is true. More importantly, if neither RNA nor IHC could be used to validate these patient subgroups, how would this ever be used in the clinic? At this point one could almost ask the authors to validate by proteomics or targeted proteomics in additional samples. It becomes complex!*

We thank the reviewer for this point, we agree that it is quite complex. We do not consider proteomics to be the universal ground truth. Of course, other biological levels such as methylation patterns have demonstrated very relevant information, e.g., in the classification of brain tumors, and gain-of-function mutations in oncogenes have longstanding evidence as predictive molecular traits. The proteome, however, is the biological level that is generally closest to the actual protein function, which is the level of relevance both in terms of tumor biology and as target of most therapeutics. When investigating these aspects, as we did, the proteome would thus be the biological level of choice. We therefore strongly believe that proteomics is not the complete picture, but that any picture will be incomplete without proteomics. In this sense, we are also convinced that proteomics is necessary to quantify targets for ADCs and other therapies accurately and reliably on a single-patient basis. While these are relatively new therapeutics (please also see our response to your point #2), we are convinced that their proven efficacy will increasingly be a mainstay of the therapy of advanced-stage cancer patients (as they are currently becoming in UC). As molecular and surgical pathologists as well as urologists, we regularly observe the lack of targetable molecular alterations by DNA or RNA in these patients, particularly in UC.

To answer your question of how this could be used in the clinic: For the short term by underlining and solidifying the relevance of luminal and basal subtypes in UC with limited classification power of transcriptomic analyses. In the long run by adding proteomics to the available techniques in molecular pathology. Our work is dedicated to making a case for this clinical application of proteomics, which is why we tried to improve the robustness of the preanalytical process (please see our response to your point #1), strictly used real-world routine diagnostic tissue, and processed all samples in analogy to the molecular pathology workflow. While the advantages of proteomics and the insufficiencies of the current standard IHC are clear to experts in the field – such as yourself – IHC is currently the only routine method for protein quantification in tissue. IHC has been a mainstay in pathology for decades, but its application is qualitative and binary, highlighting cell types by positive/negative under the conditions of each antibody and assay. This qualitative nature is preserved even in its semiquantitative scores/variants. Its linear dynamic range is narrow and shifts by antibody clone and other (unknown) preanalytical parameters (e.g., Taylor and Levenson 2006, Sompuram, Torlakovic et al. 2022). Most important, however, is its simple lack of a loading/normalization control and therefore high vulnerability to variations of preanalytical factors (very common in UC samples). We need proteomics to govern the emerging field of predictive quantitative testing for targeted therapies and we are convinced that our work adds value to this effort. We hope that our revisions of the manuscript have outlined this more clearly and thank the reviewer for addressing this important aspect.

5. *How were the UC cell lines selected for the validation of drug sensitivities.*

We selected the cell lines based on the stage and grade of the parent tumor (Zuiverloon, de Jong et al. 2018) – to cover the UC spectrum – as well as on availability and usage in the literature.

- a. *Why can the cell lines be classified into PAULA subgroups by transcriptional profiles if this only works weakly for tissues?*
- b. *4 of 7 predictions could be confirmed? This is almost random chance.*
- c. *Fig 6C hard to interpret (not very convincing).*

We thank the reviewer for these important aspects that surely needed clarification. We used transcriptional profiles, as the available data was more comprehensive (please also see the respective methods subsection) and since we had observed relatively robust classification performance with the TCGA dataset (as outlined and further solidified by your suggestions above).

At first sight, the ‘trajectory’ from our proteomic data to the CMap predictions and back to the classified cell lines is far and we understand the concerns. We are convinced, however, that given the distance between the input data and our first *in vitro* validation steps the results are even more interesting. We carefully designed the underlying data transformations with robust and conservative assumptions, e.g., by transforming all data to a quartile-like scale. These would rather lead to an underestimation of effects, but instead we found that four IC_{50} distributions in our cell line panel significantly matched the predictions. Our apologies, but we calculate the binominal probability of four false positive results (at $p=0.05$) out of seven experiments to be around 0.0002. Also, we observe very similar results in the independent drug sensitivity data from public repositories (Suppl. Fig. 10). Although it goes without saying, we should perhaps underline the fact, that we only investigated the seven compounds displayed in Fig. 6 – so there are no further, unreported negative results. Also, we feel obliged to point out that any functional validation (also in comparable publications) is limited: Often, only the functional effect of a single, correlated protein is characterized. We felt that an endpoint-driven approach (with higher risks of negative results) would map the actual biological relevance of our subtypes more comprehensively. Therefore, we tried to approximate therapeutically relevant properties, such as (differential) drug sensitivity.

Fig. 6D essentially interprets and quantifies the data from Fig. 6C. To better explain this, we revised the figure legend:

“Fig. 6: Drug repurposing and predictive correlation with the proteomic subtypes. In silico CMap predictions of (A) pathways and (B) drugs based on $n=180$ qualitative profiles of up and downregulated proteins; *: investigated in vitro; (C): In vitro dose-response curves of $n=12$ urothelial cancer cell lines, which were selected as a functional UC panel and classified within the PAULA scheme; all cell lines were treated with exemplary compounds from the in silico predictions from A; dots are single data points, line is the fitted hill-function used to calculate half-maximal inhibitory concentrations

(IC₅₀); colors as in D; (D): IC₅₀ values of the cell lines from C; in ascending order, bar base is 1 μM as indicator of sensitivity (bar downwards = more sensitive); color is predicted PAULA cluster based on RNA data, opacity is prediction confidence (cluster probability); p = (Kruskal-Wallis of sorted order, Mann-Whitney-U of quantitative values).”

To ensure proper and conservative reporting of our findings and to better communicate the underlying rationale for this type of functional validation, we additionally revised several parts of the manuscript.

In the results section:

“To further evaluate the functional and biological relevance of the proteomic subtypes, we submitted our sample-wise over- and downregulation profiles to CMap [...]

In vitro proliferation assays ~~demonstrate confirm~~ predictive correlation information with PAULA clusters

To evaluate the top representative in silico drug predictions in vitro and to investigate, whether PAULA clusters had functional/biological relevance for sensitivity towards these drugs, we selected twelve different UC cell lines as a model for functional patient samples across the UC spectrum. We evaluated the toxicity of seven different drugs, for which high cluster-specific sensitivity had been predicted, representing different molecular pathways (Fig. 6).”

In the discussion:

“We investigated whether our proteomic subtypes would correspond to specific predictions by in silico drug repurposing. Replicated CMap analyses yielded several cluster-specific pathways and compounds. We selected exemplary drugs and treated a representative cell line panel classified within the PAULA scheme to investigate if the subtypes would correlate with increased sensitivity in vitro. This way, predictions for four out of seven drugs were confirmed in vitro.”

Also, we added a statement to the limitations:

“All in vitro experiments were performed with a cell line panel of limited size and the cell lines were classified indirectly. While significant predictive correlations were observed, independent validations are necessary with higher number of replicates.”

We hope that these corrections and adjustments find the reviewer’s approval.

6. *I was not able to find the datasets in Pride using the PXD accession provided?*

We checked all three datasets using the login (<https://www.ebi.ac.uk/pride/login>) with the usernames and passwords from the methods section. We could access all three datasets this way, they are still private (not published yet accessible with the login) in the archive in line with the submission guidelines.

7. *The link to the cancerproteins.org web-interface is not very useful in its current form. It seems that only single proteins can be searched. Better functionality would be required.*

We thank the reviewer for this valuable feedback. We extended the functionality of the web page, so that the data for an unlimited number of proteins can be retrieved directly or even the complete dataset at once.

8. *The supplemental tables lack description and are formatted in a way that it is unclear what we are looking at. It's not hard to format and describe a proteomics supplemental table!*

We apologize for the suboptimal description and formatting. We had initially combined clinical and protein data into a single master file. We now revised the tables and included a legend in each file.

Overall, a potentially interesting paper that could benefit from better focussing. Some of the validations (RNA, IHC were not very convincing in current form). The same was felt for the drug sensitivities. The section on ADCs is totally oversold and at the moment not relevant.

We thank the reviewer for their interest in our paper and the very helpful suggestions and comments. We hope that the additions and changes we made to the manuscript including better explanations of the rationale for both experiments and interpretation find the reviewer’s approval.

Reviewer #2

In this article, the authors conducted a proteomic analysis of a urothelial cancer cohort consisting of 196 patients with a broad spectrum of different UC types. This study provided valuable resources for urothelial cancer research, enabling both biomarker and novel therapeutic target discovery, as well as mechanism studies. The authors developed a protocol optimized for FFPE samples. While additional benchmark analyses can be performed, the successful sample preparation protocol for proteomic analysis of FFPE samples can potentially reduce study costs and enable retrospective studies of routine diagnostic samples.

We thank the reviewer for their time, the many detailed and helpful corrections, and the appreciation of our work.

Major Comments

1. *TMT Normalization: The authors employed a custom algorithm for protein normalization and quantitation. However, the normalization algorithm neither normalizes for the total amount of material in a channel nor across TMT sets. The authors reported a predominant downregulation of proteins in tumor vs. normal samples, which could be biased by the lower total protein content in tumor vs normal samples. Failing to perform cross-plex normalization makes feature quantities from different TMT sets incomparable to each other and could affect downstream analyses, including survival and association analyses, and potentially unsupervised clustering if proper normalization wasn't applied. The authors may consider exploring TMT-integrator and PAW proteomics pipeline (https://github.com/pwilmart/PAW_pipeline). The authors could also demonstrate that their normalized protein quantitation is similar to well-recognized TMT normalization/quantitation algorithms, of which a cross-plex normalization is employed, and that biases were not introduced to their analysis.*

We thank the reviewer for addressing this very important aspect and ranking it first in their list of comments. We strongly agree that proper normalization is pivotal for any quantitative omics-scale analysis, which is why we initially included the peptide normalization aspect in our manuscript. As another reviewer felt that this aspect was too technical for the present manuscript, we adjusted Figure 2 accordingly and removed the comparison concerning the variance from the manuscript. We kept the description in the methods section, as it is not a new normalization technique (we apologize for the potentially misleading wording) but rather a comparison of two levels of normalization to a standard sample. We are grateful for the suggestion of the PAW pipeline and TMT integrator, two excellent tools, that we will surely use in the future. As we understand TMT integrator, our normalization approach is similar: We enable cross-plex comparisons by normalizing each peptide to the standard sample, which is the same in all multiplex TMT sets. We initially just showed that having this normalization step at the peptide level reduces variance. Instead of adding up all protein-grouped peptide intensities and then divide by the added intensities of the standard sample, we divided all peptides directly by their corresponding standard sample peptides. We did this to avoid the straightforward effect that different abundances (and as such different levels of variance) would lead to weighted peptides (as an extreme example one could imagine one peptide with a thousandfold higher intensity than the two others; its variance would then also be considerably higher than the absolute intensity of the other peptides; the averaging effect of having three peptides would then be diminished considerably). Of note, we had initially run the complete cluster analysis as a control on a dataset for which no peptide normalization to the standard sample (so no run/set correction) had been performed and retrieved only the different TMT sets as clusters. We did not observe any run/set effect with our actual normalization approach – neither in the distribution of TMT sets across the clusters (Fig. 3C-D), nor by PCA (Fig. 3A).

2. *TMT labeling: The article mentioned that either 5 or 10 µg samples were used depending on the total amount of peptides. This should be clarified, and it should be demonstrated that the total protein/peptide amount does not introduce bias. It was mentioned in the same paragraph that samples were labeled in random order. Are samples from the same patients always placed in the same TMT set? Cross-plex comparisons are less reliable, even with proper cross-plex TMT normalization, and it needs to be demonstrated whether downstream analysis is biased by this randomization.*

We thank the reviewer for this important concern. We had included both, TMT set type (5 or 10 µg input material) and the TMT set itself as features in PCA (Fig. 3A) and in the clustered heatmap (Fig. 3D) without evidence of bias. Also, we had included the TMT set type, initial case order and extraction order in the survival regression analysis too, in which neither parameter was a significant predictor. Concerning the TMT set composition and randomization, we had visualized the protocol in Fig. 2A but had not described it properly in the main manuscript, which we now added in the methods section and figure legend (as suggested by the reviewer, patients and healthy samples were kept in the same TMT set to avoid cross-plex comparisons):

“LC-MS/MS analyses and evaluation – TMT

TMT sets were composed in randomized order with inclusion in the same set of the paired healthy mucosa sample and other samples of the same patient, when applicable. Peptides in concatenated fractions were reconstituted in solvent A [...].”

3. *SMRD: The authors developed an protocol, SMRD, for sample preparation of FFPE samples for proteomics analysis and demonstrated its superiority over conventional sample preparation processes. It would be interesting to also compare SMRD on FFPE samples versus a conventional sample preparation method on fresh-frozen (FF) samples. The SMRD protocol involves a thermal lysis step with incubation at 90 °C for 60 min and 99 °C for 5 min. It remains unclear whether the thermal lysis introduces any artificial modifications, and it might be worthwhile to conduct an open search using MSFragger.*

This is a valuable aspect as indeed these conditions are harsh for conventional proteins. For FFPE tissue, however, these are unavoidable and necessary to ensure sufficient crosslinking reversal (please also see our response to reviewer #1 point 2a) as we and others have shown (e.g., Tian and Zhang 2010, Friedrich, Schallenberg et al. 2021, Dressler, Schoenfeld et al. 2022). We thank the reviewer for bringing MSFragger to our attention. We used standard settings for FFPE samples in our data evaluation, comparable to other FFPE studies. Concerning the comparison between FFPE and FF samples, these have been performed in the literature (e.g., Bennike, Kastaniegaard et al. 2016) showing good agreement. For our setup, this comparison was not feasible as we analyzed retrospective archival samples for which no corresponding FF tissue was available.

4. *Statistical analysis: Differential expression analysis was conducted using the non-parametric Mann-Whitney U test. However, for paired data like this, a paired t-test or a more sophisticated mixed linear model would provide greater statistical power, even after excluding patients without the normal sample. The sample-specific expression profile analysis lacks clarity, and more details are needed on how p-values are computed. The equation requires further explanation, especially regarding the meaning of B(p), and the first sentence below the equation does not make sense to me. Additionally, Figure 4C requires a more detailed explanation.*

We thank the reviewer for these important points. We have re-run our differential expression analysis with a more comprehensive dataset, also including proteins with missing values across all samples, but without missing value imputation (please also see our response to reviewer #3 point 6). We also performed a paired analysis as suggested. The results were very similar, but we had the impression that the unpaired analysis had higher power, mirrored by higher q-values and more differentially expressed proteins.

Revision Figure 6: Differential expression by pooled and paired analysis. Mann-Whitney-U and Wilcoxon test, respectively, with at least 20 % coverage across all samples or sample pairs; FDR correction by Benjamini/Hochberg in both settings; blue = higher in healthy mucosa; red = higher in tumor.

Concerning the sample-specific expression profiles, we apologize for the complicated wording. We revised the respective paragraph in the methods section:

“To fully exploit the value of our paired data, we normalized each tumor sample to its corresponding healthy mucosa (all data were \log_2 -transformed). To assign significance, we then compared each protein j of this relative sample a_i to a pseudo-individual null distribution $H(i)$. This distribution was estimated by comparison of the respective healthy sample to all other healthy samples within the same TMT set. To account for the different variances of different PSM levels, we filtered $H(i)$ with a binning function $B(p, p_j)$, which included only the part of the null distribution that contained proteins with a similar number of PSMs:

$$B(p, p_j) = \begin{cases} 1, & \text{if } \frac{1}{2}p_j < p < 2p_j \\ 0, & \text{else} \end{cases}$$

with p being the number of PSMs and p_j the number of PSMs for protein j (Fig. 4C). We computed the estimate of the standard deviation of the filtered null distribution and normalized $a_{i,j}$ with it. Using this normalized value $\tilde{a}_{i,j}$ as test statistic, we computed the p-value as $1 - \int_{-\infty}^{\tilde{a}_{i,j}} \mathcal{N}(0,1) -$ essentially the simple measure of how likely the observation of a value $\tilde{a}_{i,j}$ would be given the null distribution. [...]”

We also extended and clarified the figure legend of Fig. 4C.

5. *Figure clearance: The figures suffer from significant clarity issues. Please revise both the figures and their legends to ensure that all elements are clearly explained. It appears that '[1]' is used to indicate untransformed scales, but this notation seems redundant and causes confusion.*

We thank the reviewer for highlighting this important shortcoming of our data presentation. In line with the other comments, we have extensively revised both, figures and all figure legends. We removed the notation '[1]' throughout, which was supposed to represent the absence of a unit.

Minor Comments

1. *Page 8 paragraph 3: precursor and the fragment ion mass tolerance to 10 ppm and 0.02 assuming the unit for fragment ion mass tolerance is Da.*

The reviewer is correct, the unit was missing after 0.02, it should be “0.02 Da”. We have revised the manuscript accordingly.

2. *In the same paragraph, was contaminant proteins added to the database? How was decoy-database created?*

We used an additional fasta file composed of the 245 most common contaminant proteins (obtained from the MaxQuant software) and searched together with the human database. The contaminant protein identifications were filtered out automatically in the data analysis workflow. Proteome Discoverer always uses a decoy database search with the reversed sequences of all proteins in the databases used. The Percolator note in Proteome Discoverer calculates the false discovery rates based on the results of the decoy database search. We added this information to the methods section.

3. *Page 14 paragraph 2: What's the criteria of sufficient peptide intensities for quantitation?*

We excluded one clear outlier with the median of the non-normalized intensities 8-fold below the minimum median of all other samples. We included this information in the methods section.

4. *Page 16 paragraph 3: I assume that the pTNM groups were converted to a dummy variable or factor variable to compute an overall hazard ratio. How was that done for PAULA clusters? How was the order of clusters decided?*

That is correct. The PAULA clusters were ordered by their association with basal (more aggressive) and luminal subtypes as ends of the spectrum. The three remaining subtypes were randomly assigned to [2,3,4].

5. *Page 18 paragraph 2, "75 proteins with the highest overexpression rates across all samples": Define "overexpression rates". I assume this refers to proteins with both elevated and lowered protein abundance, and maybe a more precise term can be used. And consider using the term "protein abundance" instead of "protein expression".*

We thank the reviewer for this point, which has separately been raised by reviewer #3. We refer to our response to the respective point 5. We tried to use abundance as often as possible, the widespread use of "expression", for instance in "differential expression", and the lack of intrinsic comparative terms for abundance (overexpression != overabundance) withstanding its universal replacement.

6. *Page 20 paragraph 1 "Normalized ITGB1 intensities showed strong positive correlation with the mass spectrometric abundances" P-value and correlation method is missing.*

We added the correlation method to the figure legend. The p-value is displayed in the plot.

7. *Figure 1C: Clarify why different boxes appear to have different alpha values.*

The boxplots of the control were colored in a lighter shade.

8. *Figure 1E: Explain the meaning of "n=x/x/x/x" in the second line of each box.*

These were the replicates for each sample type (UCC #1 – NSCLC). We thank the reviewer for this point and clarified it in the figure legend.

9. *Figure 1H: Clarify the meaning of "share."*

We apologize for the wording, we meant "fraction", and corrected this throughout the manuscript.

10. *Figure 4C: Provide further clarification for this panel.*

We revised the figure legend extensively in response to your point #4.

11. *Figure 5E: Ensure that the p-value is visible and not obscured by the legends.*

We thank the reviewer for spotting this visualization error and corrected the plot.

Reviewer #3

The authors profiled the proteomes of 434 samples from 196 patients with urothelial bladder cancer. They proposed a robust proteomic classification with additional prognostic information independent from histopathological groups and showed that proteomic subtypes were predictive for these drugs by using a comprehensive cell line panel. Having said that there are significant problems with the current manuscript:

We thank the reviewer for their time and their detailed review of our work, which added many important aspects.

Major Comments

1. My major concern is with the sampling procedures. Starting from 196 patients, the authors analyzed 242 tumor samples and 192 paired healthy tissues, obtained by manually dissecting using the H&E template to yield a total sample area of ca. 1 cm². Does healthy tissue include only normal urothelium? Due to the tissues of normal urothelium being very tiny, how do the authors ensure that only normal urothelial tissues are scraped and not mixed with tumor tissue? I would like to receive some more details on this procedure and, most important, evidence should be provided that the scraped tissues correspond to the tissue annotated by the pathologists.

We thank the reviewer for addressing this pivotal point. As a matter of fact, we spent considerable time and resources on the dissection process (more than four thousand sections were processed over a period of four months, please also see our response to reviewer #1 point 2a). We followed the established procedure for dissection of specimens for molecular pathology, which we clarified in the methods section:

“One section from the middle of a series of up to ten sections was stained automatically with hematoxylin and eosin (H&E) [...] and segmented as template by a pathologist. Depending on the resulting sample area, up to ten serial sections were manually dissected using the H&E template to yield a total sample area of ca. 1 cm². [...]”

Revision Figure 7: Principal component analysis of healthy samples from the same block as the corresponding tumor sample (theoretically contaminable) and from separate blocks.

As a pathology-led study, all staff were properly trained, and the dissection process was personally overseen by the lead author (pathologist). The tissue structure and outlines can easily be merged with the template, even in unstained sections. Unclear cases were re-stained or omitted. *A priori* 627 cases were reassessed pathologically and 356 excluded due to insufficient or impure healthy tissue (please see the CONSORT diagram in Suppl. Fig. S2). We thus meticulously prepared each sample following the exact procedure that routine molecular pathology specimens are processed by every day, often also containing both healthy and tumor tissue in direct proximity (e.g., as control for microsatellite instability assays). We have additionally re-evaluated our data concerning a possible contamination with tumor tissue. We compared healthy specimens from separate tissue blocks (sections) as their corresponding tumor to healthy

specimens from the same block, the latter serving as “potentially contaminable” control. We observed no difference in their molecular profile, for instance by principal component analysis (Revision Fig. R7).

2. *How about the tumor purity of these tumor samples? Is there any difference in tumor purity among these proteomic subtypes which may influence the molecular features?*

We are thankful for this important aspect. We had prepared a supplemental figure to demonstrate the distribution of key preanalytic and clinical parameters, including tumor purity (fraction of tumor cells), which we enclose here:

Revision Figure 8 (Suppl. Fig. S3): *Key sample and processing parameters of the PAULA cohort.* (A): The fraction of tumor cells per sample stratified by proteomic PAULA clusters (violin plot and boxplot respectively, whiskers show 95 %-interval, box is interquartile range, horizontal line is mean, horizontal bar is median); (B): The unadjusted peptide concentrations in analogy to (A); (C): Relative position in the case series (of the transurethral specimens); (D): Relative position in the order the samples were extracted; (E): Patient age; (F): The proportion of transurethral samples (bar, whiskers are the 95 % confidence interval); (G): The proportion of samples in TMT high sets; (H): The proportion of male patients.

We did not observe a relevant bias, particularly not regarding the fraction of tumor cells or differences between the high and low risk clusters. Additionally, we included these parameters in the survival regression analysis (Fig. 3K-L), in which none was an independent or relevant predictor.

3. *The authors used the SMRD protocol to process samples. Is this SMRD protocol being used for the first time? Has it been reported in other published literature?*

We developed this optimized protocol specifically for our study and report it for the first time in this manuscript. We have clarified our rationale for this additional method development/reporting in our response to reviewer #1 point 1 and in the manuscript. For instance, we reworded the beginning of the results section:

“To analyze a comprehensive all-stage UC cohort from routine FFPE tissue and to measure target specificity quantitatively, we developed an optimized protocol to increase analysis depth for standard clinical samples with heterogeneous tissue types, fixation, and processing.”

We validated our protocol with three different setups and tissue from an additional cancer entity to ensure its robustness with the heterogeneous diagnostic FFPE samples and LC-MS/MS hardware. However, as the development of an ideal sample preparation protocol for diagnostic FFPE tissue is ongoing (by us and others), we have also added a respective statement to the limitations.

4. *The author obtained 242 tumor samples from 196 patients, which means that multiple tumor samples were taken from one patient. The author conducted protein clustering on the 242 tumor samples. Does this also*

mean that a patient can be classified into different protein clusters? For example, a patient can be classified in the PAULA I group with a good prognosis and also be classified in the PAULA III group with a poor prognosis. In this case, are these patients considered to have a good or poor prognosis? This is confusing and needs clarification.

We thank the reviewer for addressing this essential aspect, which surely needed clarification. We included multiple metachronous samples from some patients on purpose, to investigate how the proteomic subtypes would change over time. At each time point only one sample was included per patient, and we included only recurred or progressed samples for some patients. Suppl. Tbl. S1 lists these patients. Fig. 3I visualizes, how pTNM stages and proteomic subtypes were related over time. We had added a respective paragraph to the results section:

“Focusing on patients with multiple metachronous samples (Fig. 3I), medium and high-risk PAULA clusters preceded histopathological progression in the majority of cases. Similarly, PFS was shorter in PAULA IIa and III patients compared to PAULA I (difference of two-year restricted mean survival 3.6 and 3.9 months respectively), which was also observed in MIBC samples (two-year restricted mean survival for PAULA IIa and III 18.0 and 19.4 months vs. PAULA IIb and IIc 23.0 and 24.0 months). Suppl. Fig. S4-S6 contain all survival stratifications.”

We also created a new Suppl. Fig. S8 (Revision Figure R9) to provide more data on the temporal stability of the proteomic subtypes:

Revision Figure 9 (new Suppl. Fig. S8): *Proteomic subtypes over time in the metachronous samples.* (A): Trajectories with up to three samples per patient; color is the initial subtype; (B): Sankey plot of all pairwise transitions.

Most importantly, there was only one “downstaging” after occurrence of a high-risk subtype and within the PAULA I and II groups only few tumors progressed to the high-risk subtype. We added a sentence to the paragraph above:

“Suppl. Fig. 8 visualizes the stability of the proteomic subtypes in multiple metachronous samples from the same patients, where only a single ‘downstaging’ after the occurrence of a high-risk subtype was observed. Similarly, low-risk tumors rarely transformed to a high-risk subtype later on.”

5. *What are the criteria for the definition of overexpressed proteins? i.e. “Thymosin beta 10 (TMSB10; Fig. 4G) – the only protein also found amongst the five upregulated proteins from the standard differential expression analysis (Fig. 4A) – was overexpressed in 52 % of the >pT2 samples, and in 15 – 30 % of PAULA I, IIa, IIb, and III samples.” What is the cutoff for the definition of TMSB10 overexpressed in 52 % of the >pT2 samples? “The luminal markers SRC, RAB25, E-cadherin, and ERBB2 demonstrated only mild overexpression in PAULA I samples (up to 13 %), while the basal-associated EGFR was overexpressed in 22 % of PAULA III samples (2 % in PAULA I).” What about the mild overexpression?*

We thank the reviewer for pointing out this section, which was difficult to understand and which we have revised extensively. Overexpression rates were determined in a two-step process: First, we compared the abundance of each protein in a tumor sample to a “null-distribution” of proteins with similar numbers of unique peptides, which we obtained by comparing (dividing) the corresponding healthy tissue to all other healthy tissue samples within the TMT set. We computed an outlier statistic for each protein in a given

individual tumor sample. In the second step, we were then able to count how many tumor samples of a subset had significant over- or underexpression of a specific protein. These frequencies were reported as percentages in the manuscript. “Mild” in this sense meant “only few samples had significant overexpression”, but the wording was badly chosen and we rephrased this. We also refer to our response to reviewer #2 point 4. The beginning of this section now reads:

“[...] To fully exploit the value of our paired data, we evaluated significant over- and underexpression sample-wise by comparing each tumor sample to its individualized null distribution (Fig. 4C, see methods section for details). Counting the occurrences of significant ($p < 0.05$) over- or underexpression within a sample subset, we report these as over- and underexpression rates for an individual protein.

Fig. 4D visualizes these subset-specific expression rates of the 75 proteins with the highest general overexpression. No single protein exhibited general overexpression across all pTNM stages or PAULA subgroups.”

6. *Surprisingly, the number of upregulated proteins in tumors (5 proteins) was far less than the downregulated proteins (177 proteins) in Fig. 4A. However, tumor cells should express more proteins than normal urothelial cells to perform more functions. Should be clarified and discussed.*

This is an important aspect, and we are thankful to the reviewer for the opportunity to explain this more in depth and to revise and improve the visualizations. In the original Fig. 4A and 4B we had only analyzed proteins with full coverage across all samples. However, this was too conservative as differential expression and the resulting abundance differences can be expected to affect the identification probability itself (leading to missing values). Moreover, key regulators and signalling proteins often were of comparatively low abundance (as it can be expected from their functional role as opposed to, e.g., structural proteins). We therefore re-evaluated our data for all proteins with at least 20 % valid measurements across the samples of the respective group and updated Fig. 4A. We also replaced Fig. 4B – as it did not contain essential information – with more informative comparisons between NMIBC and MIBC samples as well as PAULA I and III samples. Concerning the healthy-vs.-tumor comparisons, this extended dataset showed considerably reduced imbalance between up- and downregulation (Rev. Fig. R10). Still, there were more proteins under- than overexpressed, which we primarily interpret in two mutually inclusive ways.

First, the heterogeneity between tumors is different from the uniformity of healthy urothelium. With a shared loss of differentiation, yet heterogeneously activated pathways and processes, proteins of the latter will be less significant statistically and their distributions will be blurred (in line with lower q-values in our analysis).

Revision Figure 10 (new Fig. 4A): Differential expression of all proteins with at least 20 % coverage across samples, x-axis are fold changes of means, y-axis are Mann-Whitney-U p-values with FDR correction by Benjamini-Hochberg.

Second, while tumor cells strongly increase proliferative pathways, they lose many regulated proteins and features, which either restrict proliferation or lead to a loss of resources. As pathologists, we constantly observe these phenomena across all entities both morphologically and in the loss of signature proteins of their tissue of origin. To evaluate this closer, we performed functional enrichment analyses of the gene ontology biological processes. Revision Fig. R11 (new Suppl. Fig. S10) lists the relevant terms. In the up-regulated group many proliferation-related processes were enriched (such as translation and metabolic processes), whereas many differentiation-related processes were found in the healthy mucosa. We added this information to the methods and results sections.

Revision Figure 11 (new Suppl. Fig. S10): *Enriched biological processes of 2997 differentially expressed proteins between healthy and tumor samples.* Ranking based on the \log_2 fold changes.

Minor Comments

7. *Page 7, paragraph 3, and paragraph 5. “The samples covered all stages of UCC...The five clusters mirrored the tumor-biological spectrum of UCC”. What is the UCC? Moreover, the authors only collected patients with urothelial bladder cancer and it is recommended to use more accurate terminology in the title of “Proteomic Analysis of the Urothelial Cancer Landscape”, as urothelial carcinomas encompass both urothelial bladder cancer and upper tract urothelial carcinoma.*

Our apologies, but we did not find the abbreviation UCC, we used UC for urothelial cancer throughout the text. Concerning the title: Yes, we analyzed bladder cancer samples, but – as pathologists – these were urothelial cancers (as opposed to squamous or adenocarcinoma). We completely agree that upper urinary tract urothelial cancers (which are a separate and distinct yet rare subset) are not covered by our study, but from our experience “urothelial cancer” is used synonymously for “urothelial carcinoma of the bladder”, whereas upper urinary tract urothelial cancers are named as such and abbreviated as UTUC. We stated this more precisely in the first sentence of the abstract and the introduction as well as in the beginning of the results and discussion sections. We also included a visualization of a urinary bladder in Fig. 1. Given this additional information, we feel that our title is not misleading. For the sake of brevity and flow, we would therefore kindly ask to keep the title as it is and hope that the reviewer will agree.

8. *Page 11, paragraph 3. “Of the seven predictions, four could be confirmed (Fig. 6E).” Where is Fig. 6E?*

Our apologies again, we referenced Fig. 6D.

9. *There are a few claims that are overstated. For example, the authors say “Normalized ITGB1 intensities showed a strong positive correlation with the mass spectrometric abundances (Fig. 5A).”, but the correlation coefficient of 0.5 would be considered only a moderate value.*

We thank the reviewer for correcting this point, we replaced strong with “moderate”. We had incorrectly used the thresholds for the effect size parameter Cramér’s V and apologize for the mix-up, which we should have noticed.

10. *The 95% confidence intervals should be presented in Fig 3A for principal component analysis.*

We have added the respective confidence ellipses in Fig. 3A and thank the reviewer for the suggestion.

11. *The method for survival analysis needs to be supplemented. For instance, what is the cutoff point for the selected sample in Figure S7?*

Suppl. Fig. S7 displays additional overall survival regressions of the non-muscle invasive samples (<pT2; subplot A) and regressions on progression-free and recurrence-free survival of all samples. We have revised the figure legend:

“Fig. S7: Categorical Cox proportional hazards regressions for further survival data and the NMIBC subset. *: $p \leq 0.05$; N.B.: the reference variables pTa low and PAULA I for the pTNM and PAULA categories are included as linear combination of the remaining categories; (A): Overall survival of the NMIBC samples (n=167), uni- (upper panel) and multivariate analysis (lower panel); (B): Recurrence-free survival of all samples; (C): Progression-free survival of all samples.”

The sample groups/subtypes were categorical, so without a cut-off (we hope that we understand this point correctly). We also refer to our comment to reviewer #2 minor point 4.

The authors thank all reviewers again for their time and their many detailed and helpful suggestions. We sincerely hope that our revised manuscript will find your approval.

References

- Adua, S. J., A. Arnal-Estape, M. Zhao, B. Qi, Z. Z. Liu, C. Kravitz, H. Hulme, N. Strittmatter, F. Lopez-Giraldez, S. Chande, A. E. Albert, M. A. Melnick, B. Hu, K. Politi, V. Chiang, N. Colclough, R. J. A. Goodwin, D. Cross, P. Smith and D. X. Nguyen (2022). "Brain metastatic outgrowth and osimertinib resistance are potentiated by RhoA in EGFR-mutant lung cancer." *Nat Commun* **13**(1): 7690.
- Bennike, T. B., K. Kastaniegaard, S. Padurariu, M. Gaihede, S. Birkelund, V. Andersen and A. Stensballe (2016). "Comparing the proteome of snap frozen, RNAlater preserved, and formalin-fixed paraffin-embedded human tissue samples." *EuPA Open Proteom* **10**: 9-18.
- Dressler, F. F., J. Schoenfeld, O. Revyakina, D. Voge, S. Kiefer, J. Kirfel, T. Gemoll and S. Perner (2022). "Systematic evaluation and optimization of protein extraction parameters in diagnostic FFPE specimens." *Clinical Proteomics* **19**(1).
- Friedrich, C., S. Schallenberg, M. Kirchner, M. Ziehm, S. Niquet, M. Haji, C. Beier, J. Neudecker, F. Klauschen and P. Mertins (2021). "Comprehensive micro-scaled proteome and phosphoproteome characterization of archived retrospective cancer repositories." *Nat Commun* **12**(1): 3576.
- Kamps, J. J. A. G., R. J. Hopkinson, C. J. Schofield and T. D. W. Claridge (2019). "How formaldehyde reacts with amino acids." *Communications Chemistry* **2**(1).
- Lee, M. C. and B. Sun (2014). "Glycopeptide capture for cell surface proteomics." *J Vis Exp*(87).
- Sompuram, S. R., E. E. Torlakovic, N. A. Hart, K. Vani and S. A. Bogen (2022). "Quantitative comparison of PD-L1 IHC assays against NIST standard reference material 1934." *Mod Pathol* **35**(3): 326-332.
- Taylor, C. R. and R. M. Levenson (2006). "Quantification of immunohistochemistry--issues concerning methods, utility and semiquantitative assessment II." *Histopathology* **49**(4): 411-424.
- Tian, Y. and H. Zhang (2010). "Isolation of proteins by heat-induced extraction from formalin-fixed, paraffin-embedded tissue and preparation of tryptic peptides for mass spectrometric analysis." *Curr Protoc Mol Biol* **Chapter 10**: Unit 10 26 11-17.
- Zuiverloon, T. C. M., F. C. de Jong, J. C. Costello and D. Theodorescu (2018). "Systematic Review: Characteristics and Preclinical Uses of Bladder Cancer Cell Lines." *Bladder Cancer* **4**(2): 169-183.

Reviewers' Comments:

Reviewer #1:

Remarks to the Author:

Overall the authors have made a solid effort to address my previous comments. While personally I would have done certain things differently, the paper is a solid resource and in-line with many other clinical proteomics papers that have been published in recent years.

As such I support publication of this manuscript in Nature Communications.

Reviewer #2:

Remarks to the Author:

Most of my previous issues have been addressed. However, after a careful review of the manuscript, I found several new minor issues.

- It's unclear how p-values in Fig 3E-H were calculated. I assume log-rank test was used (which was not mentioned), further pair-wise comparisons should be done. For example, the low p-value from Fig 3G could be dominated by two groups. And p-values should also be mentioned in the context in addition to the figures.
- Equation 2 is still confusing to me. What does S mean?
- The study cohort should be explained more clearly. It was not mentioned that metachronous samples were collected from sample patients until Fig 3I. How many patients have metachronous samples? And how long was the time interval? Same for MIBC and NMIBC that information should be given such as how they are diagnosed/defined and the fraction of them in the study cohort.
- I was not convinced by Fig 3I that "medium and high-risk PAULA clusters proceeded histopathological progression", because only 3/6 PAULA III had more progressed pTNM for their 2nd sample.
- In Fig 3E, PAULA III had shorter OS compared to PAULA IIA, but not PFS. Why?
- Top of page 13, the authors stated that low-risk tumors rarely transformed to a high-risk subtype. But I'm not very convinced by Supplementary Fig S8, because I clearly see some patients go down.
- The method to identify over- and underexpression proteins is confusing
- Page 14, 67 samples were randomly selected for immunoblots. Why not all samples?
- The consistency between protein and mRNA abundance varies by tissues and conditions (e.g., cancer types). Therefore authors should explain why it's a good idea to use the proteome data from different human tissues (Wang et al.) for filtering. Also, I think the wrong citation was used in the end of that sentence (44 instead of 66?).
- "The TCGA samples could robustly (≥ 50 % cluster probability/matching score) be re-classified for 287 out of 408 samples."

It is not immediately clear what cluster probability/matching score means. I realize that this might mean the probability that a sample belongs to a given PAULA cluster. The authors clarify it and how it is calculated as this is not mentioned in the methods section. Given that 30% patients can't be assigned to a cluster, I would be carefully describing it as "robustly".

- The PAULA clustering seems to agree with TCGA's mRNA base clustering results, and Fig 4E also showed that they have similar survival stratification. Given that tumor clustering and subtyping using genomic profile is widely accepted. What additional benefit would this proteome-based clustering method bring if it does not show much advantage?
- Page 15: "For PAULA I and IIa cell lines the tubulin inhibitors flubendazole and vincristine as well as the HDAC inhibitor apicidin showed significant relative and absolute IC50 reductions compared to PAULA III samples, indicating increased sensitivity."
P-values missing. FDR adjusted?
- Fig 6C was never cited in the paper
- Fig 3E is missing an x-axis label. I assume the authors want it to share the x-axis label with 3F, but it causes confusion.
- Fig 3D: Assuming the heatmap only includes tumor samples, but please verify. How are metachronous samples handled? Are all proteins included in the heatmap? If not, how were they filtered?
- I found Fig 3D being hard to interpret. It's unclear what role PUALA plays here. Maybe explaining

what the dots mean can help.

Reviewer #3:

Remarks to the Author:

The manuscript entitled "Proteomic Analysis of the Urothelial Cancer Landscape" performed a proteomic analysis of urothelial cancer. The authors identified 5 proteomic subtypes with prognosis information, naming PAULA clustering. However, there are significant concerns that need to be addressed, listed below.

Major Comments

- 1) The authors described that the robust proteomic subtypes with prognostic information independent from histopathological groups. However, the PAULA I subtype contained more pTa patients, and the PAULA III subtype contained more patients with pT2. There is still a lack of evidence to suggest the proteomic subtype independent from histopathological groups.
- 2) The unsupervised consensus clustering with non-negative matrix factorization (NMF) was performed on proteins of the 242 tumors, however, the 242 tumors came from 196 patients. Did this mean that the prognosis information of patients is partially duplicated?
- 3) All cell lines were classified within the PAULA scheme based on transcriptomic data. Why not use proteomic data?
- 4) The authors tried to use the NMF methods to re-classify the TCGA samples, and compared the protein markers between the two cohort. But the authors did not come to a strong conclusion.
- 5) The authors identified the proteomic clusters, and compared with TCGA cohort. However, they did not compare the proteomic clusters with other cohorts, such as MDA cohort (Choi et al.), UNC cohort (Damrauer et al.), and Fudan cohort (Xu et al.).

Minor Comments

- 1) The number of keywords in this manuscript far exceeds 5, the authors should carefully screen.
- 2) There were many grammar errors, and the manuscript often had felt convoluted and was very hard to follow.
- 3) In fig5d, there is no labeled p.value for testing between multiple groups.
- 4) In the abstract part, "We show that proteomics is relevant for personalized oncology and provide abundance and tumor specificity data for a large part of the UC proteome". I don't think it should be used the word "show".
- 5) All p value should be adjusted.

Point-by-Point Response II for Submission NCOMMS-23-41307-T

Proteomic Analysis of the Urothelial Cancer Landscape

Reviewer #1

Overall the authors have made a solid effort to address my previous comments. While personally I would have done certain things different, the paper is a solid resource and in-line with many other clinical proteomics papers that have been published in recent years.

As such I support publication of this manuscript in Nature Communications.

Reviewer #1 (Remarks on code availability): OK

We thank reviewer #1 again for their time and the appreciation of our revised work. We are very grateful for the reviewer's support that our paper should be published in *Nature Communications*.

Reviewer #2

Most of my previous issues have been addressed. However, after a careful review of the manuscript, I found several new minor issues.

We thank the reviewer for the appreciation of our revisions and the further helpful suggestions and corrections, which we addressed below.

Minor Comments

1. *It's unclear how p-values in Fig 3E-H were calculated. I assume log-rank test was used (which was not mentioned), further pair-wise comparisons should be done. For example, the low p-value from Fig 3G could be dominated by two groups. And p-values should also be mentioned in the context in addition to the figures.*

We thank the reviewer for pointing out this missing information. Indeed, log-rank tests were used, which we added to the figure legend. We have additionally computed the pairwise log-rank tests and included this information as Supplementary Data File 7 as it could not be elegantly incorporated directly in the visualization. We included a reference to these and the p-values in the text.

2. *Equation 2 is still confusing to me. What does S mean?*

We apologize for the insufficient explanation. S is the combined score for the integration of both pTNM and PAULA prognostic information. We added this to the manuscript and amended further information:

“Combining We further integrated the information from the pTNM stages (pooling \geq pT2 due to a lack of overall survival stratification and difficult histological differentiation) and PAULA clusters (Fig. 3H) by a combined score S with S_{PAULA} being the score for the PAULA clusters sorted by overall survival (ranging from 1 to 5) and S_{pTNM} being the pTNM stages as defined above (ranging from 1 to 6) [...]

This increased the HR to 4.16 (2.49 – 6.95), while the Akaike information criterion decreased from 1115/1112 to 1108.”

3. *The study cohort should be explained more clearly. It was not mentioned that metachronous samples were collected from sample patients until Fig 3I. How many patients have metachronous samples? And how long was the time interval? Same for MIBC and NMIBC that information should be given such as how they are diagnosed/defined and the fraction of them in the study cohort.*

We thank the reviewer for flagging this lack of information. We added this to the *cohort characteristics* section of the results.

“Together, n=167 samples were non-muscle-invasive (NMIBC; 69 %) and n=75 samples were muscle-invasive (MIBC; 31 %) bladder cancer specimens. Based on availability, we included two (n=34 patients) or three (n=6) metachronous samples from individual patients to study temporal effects/stability.”

4. *I was not convinced by Fig 3I that “medium and high-risk PAULA clusters proceeded histopathological progression”, because only 3/6 PAULA III had more progressed pTNM for their 2nd sample.*

This is correct, but of the other three PAULA III samples two were already MIBC and as such progressed in the clinical sense. We specified this in the respective sentence and thank the reviewer for underlining this:

“Focusing on patients with multiple metachronous samples (Fig. 3I), medium and high-risk PAULA clusters preceded histopathological progression to MIBC in most cases.”

5. *In Fig 3E, PAULA III had shorter OS compared to PAULA IIA, but not PFS. Why?*

We thank the reviewer for addressing this point. PAULA IIA tumors had a PFS similar to PAULA III samples, which we interpret as more aggressive biological behaviour compared to the other intermediate-risk PAULA subtypes (which is in turn mirrored by the genetic differences). There could be several reasons, why this

does not translate to overall survival. In general, overall survival is affected by many factors, which can blur existing differences. Most importantly, however, OS and PFS are not necessarily measures of the same biological processes, particularly not in UC with frequent, ‘small-step’ progressions and a broad spectrum of malignancy. Therefore, faster progression does not necessarily mean that these tumors are more lethal, too. Last, the response to treatment can even be better in more aggressive neoplasias (similar to high-grade lymphomas compared to indolent lymphomas). We hope that this sufficiently answers the reviewer’s question.

6. *Top of page 13, the authors stated that low-risk tumors rarely transformed to a high-risk subtype. But I’m not very convinced by Supplementary Fig S8, because I clearly see some patients go down.*

We thank the reviewer for the opportunity to explain this further. Out of the 16 PAULA I samples only two transformed to the high-risk PAULA III cluster, which we described as “rarely transformed”. The “down”-staging referred to the one PAULA III sample that had a late recurrence as a PAULA I sample, which was the only change in this subset. We hope that we understood the question correctly.

7. *The method to identify over- and underexpression proteins is confusing*

We apologize for the complicated approach and explanation. We have further revised the summary in the results section to explain this better.

8. *Page 14, 67 samples were randomly selected for immunoblots. Why not all samples?*

We thank the reviewer for the opportunity to clarify this. After SMRD processing of the samples there was only limited tissue available for many samples. In order to analyze the same dissected areas with immunoblots, a considerable amount of input tissue is needed per sample. This limited the number of available cases and, together with the resource-intensive manual processing of immunoblotting, caused the technical validation cohort to be smaller than the complete proteomics cohort.

9. *The consistency between protein and mRNA abundance varies by tissues and conditions (e.g., cancer types). Therefore authors should explain why it’s a good idea to use the proteome data from different human tissues (Wang et al.) for filtering. Also, I think the wrong citation was used in the end of that sentence (44 instead of 66?).*

We thank the reviewer for highlighting this important aspect and for spotting the reference error. We apologize for the incorrect reference; it is indeed reference 44. We corrected this and added more specific information on why we believe that this dataset is suitable:

“To classify these transcriptomic samples within our proteomic PAULA scheme, we used published data on the correlations between proteins and their mRNAs across different tissue types⁴⁴. These different biological states and cellular differentiations could be expected to represent maximum levels of biological regulation. Correlations from these samples were then likely to hold in the extrapolation to our data, too.”

10. *"The TCGA samples could robustly (≥ 50 % cluster probability/matching score) be re-classified for 287 out of 408 samples." It is not immediately clear what cluster probability/matching score means. I realize that this might mean the probability that a sample belongs to a given PAULA cluster. The authors clarify it and how it is calculated as this is not mentioned in the methods section. Given that 30% patients can’t be assigned to a cluster, I would be carefully describing it as “robustly”.*

We agree with the reviewer and thank for this comment. We removed “robustly” and rephrased the sentence. We also added the information on how this score was calculated to the methods section.

11. *The PAULA clustering seems to agree with TCGA’s mRNA base clustering results, and Fig 4E also showed that they have similar survival stratification. Given that tumor clustering and subtyping using genomic profile is widely accepted. What additional benefit would this proteome-based clustering method bring if it does not show much advantage?*

We thank the reviewer for the opportunity to elaborate on this important aspect. First, we demonstrate a basic agreement of the two biological levels. We also confirm that there are most likely five molecular subtypes in UC – the number of subtypes varied considerably between the previous transcriptomic approaches, please also see Supplementary Fig. S13. Next, we extend these subtypes to both NMIBC and MIBC, TCGA covers only MIBC. Also, there is only partial overlap, and a significant proportion of samples is classified inconsistently, particularly from the intermediate-risk PAULA subtypes. PAULA IIa shows distinct biological behavior and proteogenomic features, while it is not clearly associated with a transcriptomic subtype. Last, we underline the added benefit of using proteomics to simultaneously quantify targets to predict suitable targeted therapies.

12. *Page 15: "For PAULA I and IIa cell lines the tubulin inhibitors flubendazole and vincristine as well as the HDAC inhibitor apicidin showed significant relative and absolute IC50 reductions compared to PAULA III samples, indicating increased sensitivity." P-values missing. FDR adjusted?*

We thank the reviewer for pointing this out. We had added all p-values to Fig. 6d. These were not adjusted as we used independent cell viability assays for each compound. Across all predictions, we had previously calculated the probability of all four (out of seven) predictions being false-positives as 0.0002 in our response to reviewer #1 point 5 in the first round of revisions. This did not include our two-fold statistical approach with both parametric and non-parametric tests, which were meant to further solidify our findings.

13. *Fig 6C was never cited in the paper*

We thank the reviewer for pointing this out, we added the missing reference.

14. *Fig 3E is missing an x-axis label. I assume the authors want it to share the x-axis label with 3F, but it causes confusion.*

We thank the reviewer for spotting this suboptimal visual presentation. We added the x-axis label to Fig. 3e and update the figure.

15. *Fig 3D: Assuming the heatmap only includes tumor samples, but please verify. How are metachronous samples handled? Are all proteins included in the heatmap? If not, how were they filtered?*

We thank the reviewer for giving us the possibility to explain this further and apologize for the insufficient information in the figure legend. Fig. 3d includes all 242 tumor samples including all metachronous samples. We showed the abundances of the 25 % most cluster-relevant proteins (n=156). This information was appended to the method section and the figure legend was revised:

“(d): Abundance heatmap of all 242 tumor samples and the 25 % most cluster-relevant proteins (n=156) with clinical features and associated gene ontology biological processes; [...]”

16. *I found Fig 3D being hard to interpret. It's unclear what role PUALA plays here. Maybe explaining what the dots mean can help.*

We thank the reviewer for addressing this point and apologize for the sparse information. We assume that the reviewer is referring to Fig. 4d. We showed the individual variances of the overexpression rates for each protein from Fig. 4c to compare, whether the PAULA clusters would group these samples better (leading to more distinct differences in the rates and therefore higher variance as observed). We tried to explain this better in the figure legend and hope that our changes find the reviewer's approval.

17. *Reviewer #2 (Remarks on code availability): Page can not open*

We apologize for this problem, we have reloaded the github folder and checked it again, it should work now. Initially, we had also submitted the evaluation code separately with an upload link from the journal.

Reviewer #3

The manuscript entitled “Proteomic Analysis of the Urothelial Cancer Landscape” performed a proteomic analysis of urothelial cancer. The authors identified 5 proteomic subtypes with prognosis information, naming PAULA clustering. However, there are significant concerns that need to be addressed, listed below.

We thank the reviewer for their time, their detailed review of our manuscript, and the constructive feedback and corrections.

Major Comments

1. *The authors described that the robust proteomic subtypes with prognostic information independent from histopathological groups. However, the PAULA I subtype contained more pTa patients, and the PAULA III subtype contained more patients with pT2. There is still a lack of evidence to suggest the proteomic subtype independent from histopathological groups.*

We thank the reviewer for this important aspect. Indeed, the histopathological (pTNM) stages are associated with the PAULA clusters (which can be expected, due to the broad spectrum of our cohort and the morphological differences). We had mentioned this in the beginning of the respective paragraph of the results section:

“The five clusters mirrored the tumor-biological spectrum of UC, e.g. separating early stage pTa from muscle-invasive (MIBC) samples (Fig. 3d).“

While we further described that:

“[...] PAULA clusters contained additional prognostic information, mirrored by their survival stratification of non-muscle-invasive (NMIBC) samples (Fig. 3F) or pT1 G3 samples (Supplementary Fig. 4 [...])”

We also investigated the prognostic information of the PAULA clusters statistically, in analogy to other, comparable publications (e.g. Xu, Yao et al. (2022)). Here, PAULA clusters contained independent information in the multivariate Cox proportional hazards regression, adjusting for the pTNM stages, age, and tumor purity (the significant parameters from the univariate analysis; Fig. 3I; hazard ratio 1.19 (95%-CI 1.03-1.38); $p=0.017$; comparable to the pTNM hazard ratio of 1.22 (95%-CI 1.09-1.37)). Also, the combination of both pTNM and PAULA subgroups by Eq. 2 increased the hazard ratio considerably to 4.16 (2.49-6.95) (Fig. 3h).

2. *The unsupervised consensus clustering with non-negative matrix factorization (NMF) was performed on proteins of the 242 tumors, however, the 242 tumors came from 196 patients. Did this mean that the prognosis information of patients is partially duplicated?*

We thank the reviewer for this relevant aspect that surely needed better communication. We included (for a small subset, please see also our response to reviewer #2 point 3) up to two metachronous samples per patient, to study the stability and prognostic power of our subtypes over time. For the survival analyses in Fig. 3 we included each of these samples as independent tumor, but the prognostic data was not duplicated. Instead, the respective follow-up/survival timespans were accurately calculated for each individual sample. Of note, only 20 % of the patients had more than one sample, of those 34 had two and 6 patients had three samples. We have additionally repeated our analyses with only a single sample per patient (the initial tumor in our series) as Revision Fig. 12. These comparisons for both overall and progression-free survival do not indicate relevant differences. We included this information as new Supplementary Fig. 4.

Revision Figure 12 (new Supplementary Fig. 4): *Comparability of multiple metachronous samples per patient versus single samples.* Kaplan-Meier survival curves for all tumor samples (left) or only single samples per patient (initial tumor; right).

3. *All cell lines were classified within the PAULA scheme based on transcriptomic data. Why not use proteomic data?*

We agree with the reviewer that this would have been ideal. However, we used available transcriptomic data as there was no proteomic data available for all the cell lines we investigated *in vitro*. Also, this approach had worked reasonably well for the TCGA samples (and did so for the cell lines, too). Nonetheless, it remains a limitation of our study, which we have addressed in the limitations paragraph at the end of the discussion:

“Last, all *in vitro* experiments were performed with a cell line panel of limited size and the cell lines were classified indirectly. While significant predictive correlations were observed, independent validations are necessary with higher number of replicates.”

4. *The authors tried to use the NMF methods to re-classify the TCGA samples, and compared the protein markers between the two cohort. But the authors did not come to a strong conclusion.*

We thank the reviewer for addressing this important aspect. Our apologies if the manuscript was not clear, but we followed a slightly different approach. We used support vector classification to re-classify the TCGA samples based on a thoroughly filtered and conservatively transformed subset of proteins and their (correlated) transcripts. We did this to evaluate if and how the proteomic subtypes would correlate with the transcriptomic TCGA subtypes. These analyses, however, were used in a validation and contextualization sense and not as a primary objective, which is why we did not draw a single ‘strong’ conclusion but rather focused on a variety of relevant aspects.

We described these results in the section *Proteogenomic context with the TCGA dataset*, for instance:

“We observed a general correlation of luminal (papillary) and basal (squamous) subtypes with PAULA I and III samples, respectively, while PAULA II subtypes were more heterogeneously associated (Fig. 5D, Supplementary Fig. 12).”

We further used the associated survival data of the TCGA cohort to successfully validate the prognostic value of our proteomic subtypes (Fig. 5e). Additionally, we investigated the associated genomic and epigenetic data and reported key findings:

“Interestingly almost all *FGFR3* mutations (which are a hallmark of luminal tumors) were contained in the PAULA I samples, whereas they were found more evenly distributed across luminal papillary, luminal, and neuronal subtypes in the TCGA data (similar frequencies were found in basal squamous and PAULA III samples). *TP53* mutations in turn were enriched in both PAULA III and PAULA IIa samples, two clusters that also showed subtle enrichment of hypomethylation clusters 4 and 1, respectively. Copy number variations such as amplifications in *PPARG*, *FGFR1*, and *NECTIN4* were more frequently observed in PAULA IIa samples, while neither *CDKN2A* (p16; as proposed marker for genomically unstable subtypes) gene alterations nor protein abundance (Supplementary Fig. 9B) showed an association. Interestingly, the higher *NECTIN4* amplification rates in PAULA IIa samples were mirrored by higher protein abundances in PAULA IIa samples in our cohort (Supplementary Fig. 11).”

We integrated these findings, which validated our proteomic data, in different paragraphs of the discussion, for instance:

“Interestingly, PAULA IIa samples demonstrated increased abundance of Nectin-4 and were associated with *NECTIN4* amplifications in the re-classified TCGA samples.” (p.18)

We thank the reviewer again for highlighting that the use of the TCGA dataset as external validation was not clear enough in the manuscript. We therefore modified the respective main statement in the discussion section (p.16):

“We used further, re-classified MIBC samples of the TCGA cohort as external validation of the prognostic value of the PAULA subtypes. Here, as external validation set, PAULA clusters also demonstrated relevant OS stratification.”

5. The authors identified the proteomic clusters, and compared with TCGA cohort. However, they did not compare the proteomic clusters with other cohorts, such as MDA cohort (Choi et al.), UNC cohort (Damrauer et al.), and Fudan cohort (Xu et al.).

We thank the reviewer for suggesting this extended context. We decided to use the TCGA cohort for our primary comparisons in the main manuscript as it is the most comprehensive dataset with the deepest molecular multi-level characterization. It is widely considered the gold standard of molecular classifications and as such included in the WHO classification of urothelial cancer (IARC 2022). We have compared our results with the Xu et al. publication in the discussion (p.16). Concerning the other molecular (transcriptomic) classifications mentioned, we have shown their overlap with our proteomic subtypes in Supplementary Figure S13 (Revision Fig. R13):

Revision Figure 13 (Supplementary Fig. 13): *Overlap with transcriptomic classifications.* Sankey plots of the proteomic (left) and respective transcriptomic clusters (right) based on all reclassified TCGA samples; Based on the supplemental data from Robertson, Kim et al. (2017; TCGA 2017); TCGA 2014: Cancer Genome Atlas Research (2014); CIT-Curie: Rebouissou, Bernard-Pierrot et al. (2014); UNC: Damrauer, Hoadley et al. (2014); MDA: Choi, Porten et al. (2014); Lund: Sjobahl, Lauss et al. (2012).

Minor Comments

1. *The number of keywords in this manuscript far exceeds 5, the authors should carefully screen.*

We thank the reviewer for pointing this out. We have limited the keywords to five and removed duplicated or similar terms.

~~“Urothelial cancer; Protein extraction; Clinical proteomics; Risk stratification; Personalized medicine; Bladder cancer; Molecular classification; Pathology; Urological oncology; Urology; Oncology; Drug prediction; Proteomics; Molecular oncology; Diagnostics; Personalized oncology; Translational pathology; Predictive testing; Antibody-drug conjugates”~~

2. *There were many grammar errors, and the manuscript often had felt convoluted and was very hard to follow.*

We thank the reviewer for this suggestion, we carefully revised the manuscript with a special focus on grammar and wording.

3. *In fig5d, there is no labeled p.value for testing between multiple groups.*

We thank the reviewer for pointing this out. We added the respective comparisons as Supplementary Data 7 as we did not find a good way to include these directly in the visualization. These data include all unadjusted and FDR-corrected comparisons. We also added this information to the manuscript.

4. *In the abstract part, “We show that proteomics is relevant for personalized oncology and provide abundance and tumor specificity data for a large part of the UC proteome”. I don’t think it should be used the word “show”.*

We thank the reviewer for this suggestion and adjusted the abstract accordingly by replacing “show” with “underline”, which indeed captures the essence of our work better than the previous wording. The sentence now reads:

“We underline show that proteomics is relevant for personalized oncology and provide abundance and tumor specificity data for a large part of the UC proteome (www.cancerproteins.org).”

5. *All p value should be adjusted.*

We thank the reviewer for highlighting this point. We adjusted all p-values where multiple comparisons were performed.

6. *Reviewer #3 (Remarks on code availability): We tried to access the websites, and cannot open.*

We apologize for this problem, we have reloaded the github folder and checked it again, it should work now. Initially, we had also submitted the evaluation code separately with an upload link from the journal.

The authors thank all reviewers again for their time and their many detailed and helpful suggestions. We sincerely hope that our revised manuscript will find your approval.

References

- Cancer Genome Atlas Research, N. (2014). "Comprehensive molecular characterization of urothelial bladder carcinoma." *Nature* **507**(7492): 315-322.
- Choi, W., S. Porten, S. Kim, D. Willis, E. R. Plimack, J. Hoffman-Censits, B. Roth, T. Cheng, M. Tran, I. L. Lee, J. Melquist, J. Bondaruk, T. Majewski, S. Zhang, S. Pretzsch, K. Baggerly, A. Siefker-Radtke, B. Czerniak, C. P. Dinney and D. J. McConkey (2014). "Identification of distinct basal and luminal subtypes of muscle-invasive bladder cancer with different sensitivities to frontline chemotherapy." *Cancer Cell* **25**(2): 152-165.
- Damrauer, J. S., K. A. Hoadley, D. D. Chism, C. Fan, C. J. Tiganelli, S. E. Wobker, J. J. Yeh, M. I. Milowsky, G. Iyer, J. S. Parker and W. Y. Kim (2014). "Intrinsic subtypes of high-grade bladder cancer reflect the hallmarks of breast cancer biology." *Proc Natl Acad Sci U S A* **111**(8): 3110-3115.
- IARC (2022). *Urinary and Male Genital Tumours*. Lyon, IARC.
- Rebouissou, S., I. Bernard-Pierrot, A. de Reynies, M. L. Lepage, C. Krucker, E. Chapeaublanc, A. Herault, A. Kamoun, A. Caillault, E. Letouze, N. Elarouci, Y. Neuzillet, Y. Denoux, V. Molinie, D. Vordos, A. Laplanche, P. Maille, P. Soyeux, K. Ofualuka, F. Reyat, A. Biton, M. Sibony, X. Paoletti, J. Southgate, S. Benhamou, T. Le Bret, Y. Allory and F. Radvanyi (2014). "EGFR as a potential therapeutic target for a subset of muscle-invasive bladder cancers presenting a basal-like phenotype." *Sci Transl Med* **6**(244): 244ra291.
- Robertson, A. G., J. Kim, H. Al-Ahmadie, J. Bellmunt, G. Guo, A. D. Cherniack, T. Hinoue, P. W. Laird, K. A. Hoadley, R. Akbani, M. A. A. Castro, E. A. Gibb, R. S. Kanchi, D. A. Gordenin, S. A. Shukla, F. Sanchez-Vega, D. E. Hansel, B. A. Czerniak, V. E. Reuter, X. Su, B. de Sa Carvalho, V. S. Chagas, K. L. Mungall, S. Sadeghi, C. S. Peadarallu, Y. Lu, L. J. Klimczak, J. Zhang, C. Choo, A. I. Ojesina, S. Bullman, K. M. Leraas, T. M. Lichtenberg, C. J. Wu, N. Schultz, G. Getz, M. Meyerson, G. B. Mills, D. J. McConkey, T. R. Network, J. N. Weinstein, D. J. Kwiatkowski and S. P. Lerner (2017). "Comprehensive Molecular Characterization of Muscle-Invasive Bladder Cancer." *Cell* **171**(3): 540-556 e525.
- Sjodahl, G., M. Lauss, K. Lovgren, G. Chebil, S. Gudjonsson, S. Veerla, O. Patschan, M. Aine, M. Ferno, M. Ringner, W. Mansson, F. Liedberg, D. Lindgren and M. Hoglund (2012). "A molecular taxonomy for urothelial carcinoma." *Clin Cancer Res* **18**(12): 3377-3386.
- Xu, N., Z. Yao, G. Shang, D. Ye, H. Wang, H. Zhang, Y. Qu, F. Xu, Y. Wang, Z. Qin, J. Zhu, F. Zhang, J. Feng, S. Tian, Y. Liu, J. Zhao, J. Hou, J. Guo, Y. Hou and C. Ding (2022). "Integrated proteogenomic characterization of urothelial carcinoma of the bladder." *J Hematol Oncol* **15**(1): 76.